# The KU-PARP14 axis differentially regulates DNA resection at stalled replication forks by MRE11 and EXO1

Ashna Dhoonmoon[1], Claudia M. Nicolae [1]✉ & George-Lucian Moldovan [1]✉

Suppression of nascent DNA degradation has emerged as an essential role of the BRCA pathway in genome protection. In BRCA-deficient cells, the MRE11 nuclease is responsible for both resection of reversed replication forks, and accumulation of single stranded DNA gaps behind forks. Here, we show that the mono-ADP-ribosyltransferase PARP14 is a critical co-factor of MRE11. PARP14 is recruited to nascent DNA upon replication stress in BRCA-deficient cells, and through its catalytic activity, mediates the engagement of MRE11. Loss or inhibition of PARP14 suppresses MRE11-mediated fork degradation and gap accumulation, and promotes genome stability and chemoresistance of BRCA-deficient cells. Moreover, we show that the KU complex binds reversed forks and protects them against EXO1-catalyzed degradation. KU recruits the PARP14-MRE11 complex, which initiates partial resection to release KU and allow long-range resection by EXO1. Our work identifies a multistep process of nascent DNA processing at stalled replication forks in BRCA-deficient cells.

ADP ribosylation, which involves the reversible addition of ADP-ribose to substrates, has emerged as a post-translational modification which controls a variety of biological processes[1–3]. ADP ribosylation is catalyzed by ADP-ribosyltransferases of the PARP family. The best-studied enzyme of this group is PARP1, which catalyzes formation of poly-ADP-ribose (PAR) chain formation. PARP1 has multiple cellular roles including in DNA base excision repair and DNA damage signaling, by catalyzing poly-ADP-ribosylation (PARylation) at sites of DNA damage[4]. PARP1 inhibition is particularly toxic in certain genetic backgrounds with compromised DNA repair, such as mutations in the BRCA pathway, and PARP1 specific inhibitors are successfully used in the treatment of BRCA-mutant ovarian and breast tumors[5,6].

A subset of PARP family members have a specific change in the catalytic PARP domain which blocks their ability to catalyze PAR chain formation, and thus are only able to transfer a single ADP-ribose molecule unto the substrate, a process termed mono-ADP-ribosylation (MARylation)[1–3]. The cellular functions and substrates of MARylation are much less understood compared to those of PARP1-catalyzed PARylation. One such mono-ADP-ribosyltransferase is PARP14, which was been initially described as a transcription co-factor in macrophages[7–9], and has subsequently been shown to participate in multiple signal transduction pathways[10–14], as well as to regulate mRNA stability[7,15]. Unbiased proteomic-based studies have identified over a hundred potential PARP14 substrates, with functions varying from metabolism to protein translation to DNA repair[15], but how exactly MARylation by PARP14 impacts the functions of these substrates is not known. Specific inhibitors of PARP14 catalytic activity have recently been developed, and targeting PARP14 has been proposed as a possible therapeutic approach for multiple cancer types including pancreatic, hepatic and multiple melanoma, since pro-survival/proliferation and anti-apoptotic activities of PARP14 have been described in these cancers[10,13,14,16–18].

We have previously identified a role for PARP14 in the cellular response to replication stress. We found that PARP14-deficient cells were mildly impaired in maintaining DNA replication rates in response to replication-specific DNA damaging agents such as camptothecin[19]. We furthermore described a synthetic lethal interaction between PARP14 and the replication stress checkpoint kinase ATR, and identified altered DNA replication fork dynamics as the underlying factor for this genetic interaction[20]. The mechanism

[1]Department of Biochemistry and Molecular Biology, The Pennsylvania State University College of Medicine, Hershey, PA 17033, USA.
✉e-mail: cmn14@psu.edu; glm29@psu.edu

employed by PARP14 to control the response of replication forks to replication stress remains unclear.

Upon replication stress, ongoing replication forks decelerate and may eventually arrest[21]. At sites of DNA lesions, arrested forks can undergo fork reversal, by annealing the nascent strands of the sister chromatids[21–23]. This allows the processing of arrested forks for eventual fork restart, and avoids prolonged replication fork stalling, which could otherwise lead to fork breakage and collapse, generating genomic instability[24,25]. The processing of reversed forks may involve controlled resection by DNA nucleases, in particular MRE11[26–28]. The BRCA pathway, while canonically viewed as an essential component of the Homologous Recombination (HR) DNA repair machinery, has been shown to regulate DNA resection by MRE11 at reversed replication forks. In BRCA-deficient cells, nascent DNA is nucleolytically degraded by MRE11[29], in a process subsequent to fork reversal[28,30]. In addition to MRE11 inhibition, inactivation of DNA nucleases CTIP, EXO1 or DNA2 also suppresses, to a certain extent, stalled fork degradation in BRCA-deficient cells[27,31], but how these nucleases are coordinated to perform fork degradation under these circumstances is not completely understood. Nevertheless, degradation of replication forks in BRCA-deficient cells is an important component of genomic instability in these cells, and protection of stalled forks against nucleolytic degradation is associated with chemoresistance of BRCA-mutant tumors[22,23,30–33]. In addition, a recently discovered function of the BRCA pathway in suppressing ssDNA gap formation also involves inhibition of MRE11 activity, and may regulate the sensitivity of BRCA-deficient cells to PARP1 inhibitors[34–40]. These findings highlight the translational relevance of controlling the nucleolytic degradation of nascent DNA during DNA replication.

Double stranded DNA break (DSB) ends, including those of single-ended DSBs formed upon replication fork collapse, are protected by immediate loading of the KU complex, a heterodimer composed of the KU70 (XRCC6) and KU80 (XRCC5) proteins[41–43], which stabilizes the DSB end structure, and inhibits exonucleolytic degradation by EXO1. Upon binding to DSB ends, KU recruits NHEJ components to perform the DNA end ligation reaction. For the DSB to be repaired by HR instead of NHEJ, the KU complex needs to be removed from DSB ends, to allow DSB end resection. This occurs through the endonuclease activity of MRE11, activated by its cofactor CTIP, which creates a single-stranded nick proximal to KU. This serves as an entry point for EXO1-mediated long-range resection, while at the same time allowing MRE11 exonuclease activity to process the remaining short strand from the nick towards the KU-bound end (short-range resection), resulting in KU removal[44–53]. Recent work in yeast indicated that KU also binds to stalled replication forks in the absence of fork collapse and DSB formation, suggesting that KU may bind to the one-ended DSB end exposed upon fork reversal; It was proposed that, similar to DSB processing during DSB repair, MRE11 mediates the removal of KU from reversed forks, and this results in long-range resection by EXO1[54,55]. Whether KU binds reversed replication forks in higher eukaryotes, and the impact of this binding on fork degradation in BRCA-deficient cells, is unclear.

Here, we show that PARP14 interacts with MRE11, and, through its catalytic MARylation activity, fosters MRE11 engagement on nascent DNA and subsequent genomic instability and DNA damage hypersensitivity in BRCA-deficient cells. PARP14 inactivation suppresses MRE11-dependent fork degradation and ssDNA gap accumulation, and promotes chemoresistance of BRCA-mutant cells. We moreover show that the KU complex binds reversed replication forks in human cells, and mediates the initial processing of these structures by PARP14-MRE11. In BRCA-deficient cells, this releases uncontrolled resection by EXO1. Our work identifies PARP14 and KU as essential components of the machinery processing stalled replication forks, and delineates a multi-step mechanism for this processing, involving the consecutive engagement of multiple nucleases.

## Results

### PARP14 regulates fork degradation in BRCA-deficient cells

Protection of stalled replication forks against nucleolytic degradation is an essential activity of the BRCA pathway in genomic stability. Since we previously identified a role for PARP14 in replication fork dynamics[19,20], we sought to investigate if PARP14 impacts fork degradation in BRCA-deficient cells. As previously described[29,56], HU treatment induced degradation of nascent DNA in BRCA1- and BRCA2-deficient cells, as measured by DNA fiber combing. In line with previous reports[57,58], we found that this degradation is dependent on both the endonuclease and the exonuclease activities of MRE11, since it can be suppressed by both the MRE11 endonuclease inhibitor PFM01, as well as the MRE11 exonuclease inhibitor mirin (Supplementary Fig. 1a–c). While knockdown of PARP14 in wildtype cells did not affect fork degradation, we found that PARP14 depletion in HeLa BRCA2-knockout (HeLa-BRCA2$^{KO}$) cells, using multiple separate siRNA oligonucleotides, suppresses HU-induced fork degradation in these cells, similar to MRE11 inhibition (Fig. 1a, b; Supplementary Fig. 1d–f). Unlike PARP14 depletion, knockdown of the related mono-ADP-ribosyltransferase PARP10 did not suppress fork degradation (Supplementary Fig. 1g, h). To rule out potential non-specific effects of the siRNA treatment, we employed CRISPR/Cas9 to knock-out PARP14 in HeLa-BRCA2$^{KO}$ cells. We obtained two independent double-knockout clones (Supplementary Fig. 1i). Both HeLa-BRCA2$^{KO}$PARP14$^{KO}$ clones showed suppression of HU-induced fork degradation compared to HeLa-BRCA2$^{KO}$ single knockout clones (Fig. 1c). Altogether, these results demonstrate a specific impact of PARP14 on fork degradation in BRCA2-deficient cells.

Degradation of stalled replication forks requires their reversal, mediated by DNA translocases including HLTF, SMARCAL1 and ZRANB3, which catalyze the annealing of the nascent strands of the two sister chromatids to each other[21–23,28,30]. As previously shown, depletion of ZRANB3 suppressed fork degradation in BRCA2-deficient cells (Supplementary Fig. 1g, h), since fork reversal is defective under these conditions. We thus sought to test if the impact of PARP14 on fork degradation reflects a role for PARP14 in fork reversal. To this end, we employed the DNA fiber combing assay to measure fork slowing upon treatment with a low dose of HU, which is a surrogate readout for fork reversal[59]. Unlike ZRANB3 depletion, which as expected suppressed fork slowing, depletion of PARP14 had no impact (Fig. 1d; Supplementary Fig. 1j). These findings argue against a role for PARP14 in fork reversal. To confirm this using an different approach, we employed the SIRF (in situ detection of proteins at replication forks) assay, a proximity ligation (PLA)-based assay which measures binding of proteins to nascent DNA[60], to measure PARP1 engagement on stalled replication forks. Since PARP1 is a critical regulator of fork reversal[61], the presence of PARP1 on nascent DNA upon replication stress has been previously used as a readout for fork reversal[62]. Unlike ZRANB3 depletion, which as expected reduced PARP1 SIRF foci, PARP14 knockdown did not affect it (Fig. 1e). Overall, these experiments suggest that PARP14 is not involved in fork reversal.

We next sought to investigate if the impact of PARP14 loss on fork degradation is specific to HeLa-BRCA2$^{KO}$ cells. PARP14 depletion also suppressed HU-induced fork degradation in DLD1-BRCA2$^{KO}$ cells (Fig. 1f; Supplementary Fig. 1k). Moreover, we observed a similar suppression in RPE1-BRCA1$^{KO}$ cells (Fig. 1g; Supplementary Fig. 1l), demonstrating that the effect of PARP14 depletion is not cell line specific, and occurs in both BRCA1 and BRCA2-deficient cells. In line with this, PARP14 depletion also suppressed fork degradation induced in wildtype HeLa cells by treatment with the RAD51 inhibitor B02 (Fig. 1h), which mimics BRCA pathway deficiency and was previously shown to induce MRE11-mediated nascent strand resection in BRCA-proficient cells[30].

Finally, we employed a PARP14 genetic complementation system. We previously obtained multiple PARP14-knockout clones in 8988 T cells, and complemented one of these clones by stable exogenous

 

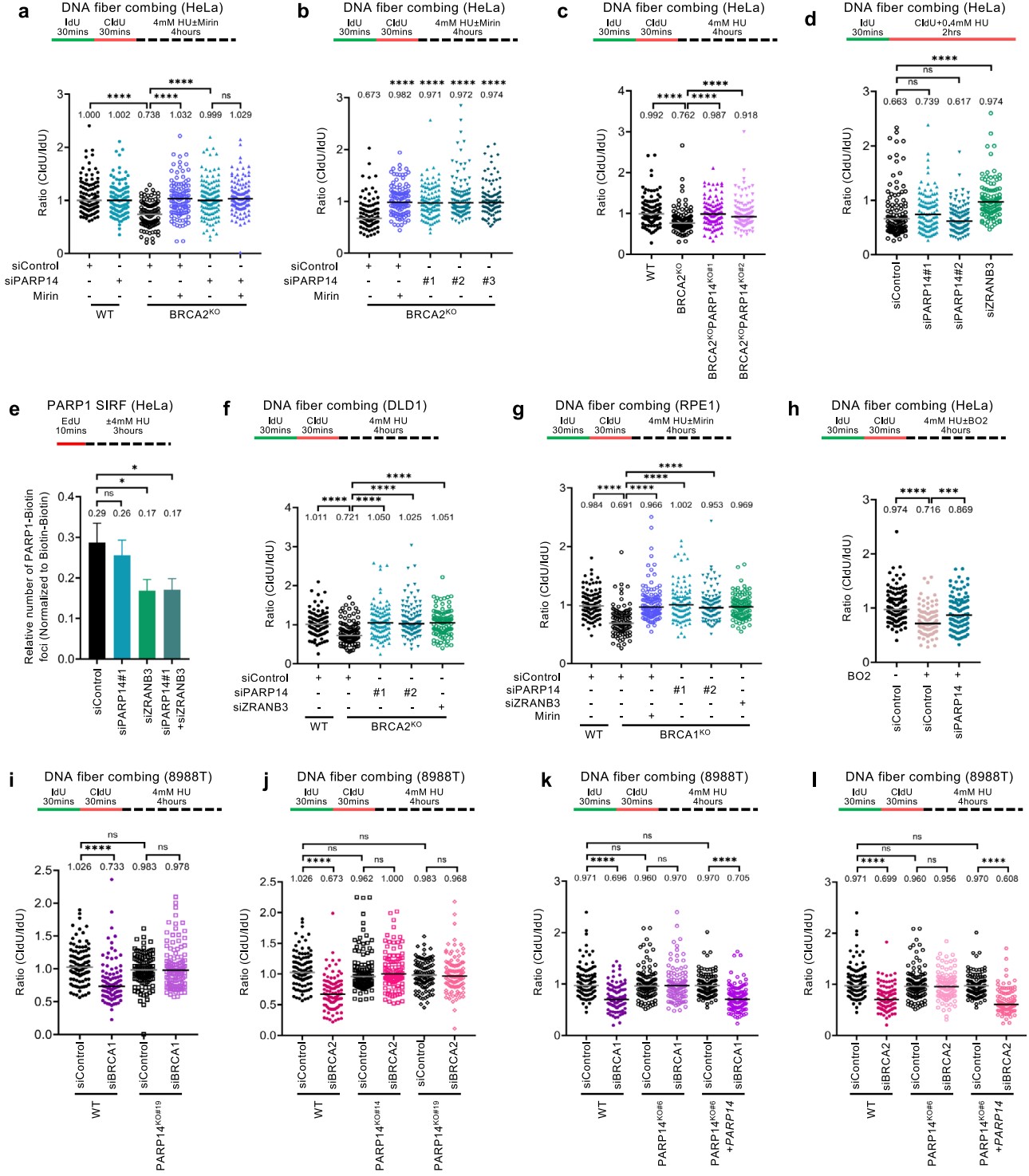

re-expression of PARP14 cDNA[20]. Knockdown of BRCA1 or BRCA2 resulted in HU-induced fork degradation in wildtype 8988 T cells, but not in PARP14-knockout 8988 T cells (Fig. 1i, j; Supplementary Fig. 1m). Moreover, HU-induced fork degradation was re-established upon BRCA1 or BRCA2 depletion in PARP14-knockout 8988 T cells corrected by re-expression of PARP14 cDNA (Fig. 1k, l). Overall, these findings show that loss of PARP14 restores fork protection to BRCA-deficient cells.

## PARP14 mediates genomic instability in BRCA-deficient cells

Fork degradation causes genomic instability, and fork protection was previously associated with chemoresistance of BRCA-deficient

cells[22,23,30–33]. Thus, we next tested if loss of PARP14 promotes genomic stability in these cells. Strikingly, PARP14 depletion reduced γH2AX foci in HU-treated BRCA2-knockout cells down to wildtype levels (Fig. 2a; Supplementary Fig. 2a), suggesting that HU-induced DNA damage is suppressed by PARP14 loss in these cells. To test if the suppression of DNA damage by PARP14 loss was specific to HU, we employed the neutral comet assay to measure DNA damage accumulation upon treatment with the chemotherapeutic agents cisplatin and olaparib (a PARP1 inhibitor). Treatment of BRCA2-knockout HeLa cells with these agents drastically induced DNA damage compared to control cells, but this increase was completely suppressed upon PARP14 depletion (Fig. 2b, c; Supplementary Fig. 2b). Similar results were

**Fig. 1 | Loss of PARP14 promotes replication fork protection in BRCA-deficient cells. a, b** DNA fiber combing assays showing that PARP14 knockdown, with three different siRNA oligonucleotides, suppresses HU-induced nascent strand degradation in HeLa-BRCA2[KO] cells. Treatment with the MRE11 inhibitor mirin is used as control. Western blots confirming the PARP14 knockdown are shown in Supplementary Fig. 1d. **c** DNA fiber combing assay showing that HeLa-BRCA2[KO]PARP14[KO] double knockout cells show restoration of fork protection upon HU treatment compared to HeLa-BRCA2[KO] cells. Two independent clones were investigated. Western blots confirming the PARP14 and BRCA2 knockout are shown in Supplementary Fig. 1i. **d** DNA fiber combing assays showing that PARP14 knockdown does not affect HU-induced fork slowing in HeLa cells. Knockdown of the DNA translocase ZRANB3, which is essential for fork reversal, is used as a control for defective fork slowing. **e** PARP1 SIRF experiment showing that PARP14 depletion does not affect PARP1 binding to nascent DNA. At least 100 cells were quantified for each condition. Bars indicate the mean values, error bars represent standard error of the mean, and asterisks indicate statistical significance (t-test, two-tailed, unpaired). A schematic representation of the assay conditions is shown at the top. **f, g** DNA fiber combing assays showing that PARP14 knockdown suppresses HU-induced nascent strand degradation in DLD1-BRCA2[KO] (**f**) and RPE1-BRCA1[KO] (**g**) cells. Treatment with the MRE11 inhibitor mirin, or knockdown of the ZRANB3 translocase, are used as control. **h** DNA fiber combing assay showing that PARP14 knockdown suppresses HU-induced nascent strand degradation in wildtype HeLa cells treated with the RAD51 inhibitor B02. **i, j** DNA fiber combing assays showing that depletion of BRCA1 (**i**) or BRCA2 (**j**) causes HU-induced fork degradation in wildtype, but not in PARP14-knockout 8988 T cells. Western blots confirming the BRCA1 and BRCA2 knockdown are shown in Supplementary Fig. 1m. **k, l** DNA fiber combing assays showing that complementation of 8988T-PARP14[KO] cells by stable re-expression of PARP14 cDNA restores HU-induced fork degradation upon depletion of BRCA1 (**k**) or BRCA2 (**l**). For all DNA fiber combing panels, the ratio of CldU to IdU tract lengths is presented, with the median values marked on the graphs and listed at the top. At least 100 tracts were quantified for each sample. Asterisks indicate statistical significance (Mann-Whitney, two-tailed). Schematic representations of the assay conditions are shown at the top. Source data are provided as a Source Data file.

obtained in BRCA1-knockout RPE1 cells (Fig. 2d, e). Overall, these results indicate that PARP14 promotes genomic instability in BRCA-deficient cells upon treatment with DNA damaging chemotherapeutic agents.

We next tested if this PARP14-dependent genomic instability mediates the hypersensitivity of BRCA-deficient cells to these agents. PARP14 depletion significantly suppressed the olaparib and cisplatin hypersensitivity of HeLa-BRCA2[KO] (Fig. 2f, g) and DLD1-BRCA2[KO] (Fig. 2h, i) cells. Moreover, the double-knockout HeLa-BRCA2[KO]PARP14[KO] cells were significantly less sensitive to olaparib and cisplatin than HeLa-BRCA2[KO] single knockout cells (Fig. 2j, k; Supplementary Fig. 2c, d). In addition, PARP14 depletion suppressed olaparib and cisplatin-induced apoptosis in RPE1-BRCA1[KO] cells (Supplementary Fig. 2e, f), as well as the cisplatin sensitivity of MDA-MB-436 patient-derived BRCA1-mutant breast cancer cells (Supplementary Fig. 2g). Finally, BRCA1- or BRCA2-depleted PARP14-knockout 8988 T cells were less sensitive to olaparib or cisplatin, compared to BRCA1- or BRCA2-depleted wildtype 8988 T cells (Fig. 2l, m). Overall, these findings indicate that loss of PARP14 promotes chemoresistance of BRCA1/2-deficient cells. To assess if these findings are clinically relevant, we analyzed publicly available TCGA datasets of breast invasive carcinoma[63] and found that PARP14 expression can stratify the survival of breast cancer patients with BRCA2-mutant tumors: low PARP14 levels are associated with reduced survival (Supplementary Fig. 2h), in line with the idea that PARP14 deficiency promotes chemoresistance in BRCA2-mutant cells.

## PARP14 promotes MRE11 engagement on nascent DNA
We next sought to investigate how PARP14 promotes genomic instability and chemosensitivity in BRCA-deficient cells. A previously-described mechanism of chemoresistance in these cells is restoration of homologous recombination[64]. We thus employed the DR-GFP reporter assay in U2OS cells[65], to test the impact of PARP14 loss on HR in BRCA-deficient cells. PARP14 knockdown in wildtype cells led to a small decrease in HR efficiency, as we previously documented[19]. However, its co-depletion in BRCA1- or BRCA2-knockdown cells did not impact the HR reduction conferred by depletion of BRCA1 or BRCA2 (Supplementary Fig. 3a–d). Moreover, PARP14 depletion did not restore RAD51 foci upon camptothecin treatment in BRCA2-knockout HeLa cells (Supplementary Fig. 3e). Overall, these findings suggest that chemoresistance by PARP14 loss does not involve restoration of RAD51-mediated HR in BRCA-deficient cells.

Recently, suppression of ssDNA gap accumulation during DNA replication has been identified as a novel activity of the BRCA pathway; ssDNA gap accumulation in BRCA-deficient cells is associated with genomic instability, and gap suppression promotes chemoresistance in these cells[34–40]. We thus sought to test if genomic stability conferred

by PARP14 loss in BRCA-deficient cells involves gap suppression. We employed the BrdU alkaline comet assay to measure replication-associated gap formation upon treatment with a low dose of HU to induce mild replication stress, as previously described[40,66]. In line with the previously-documented role of the BRCA pathway in gap suppression, BRCA2-knockout HeLa cells accumulated ssDNA gaps upon HU treatment; However, PARP14 depletion in these cells suppressed gap accumulation (Fig. 3a; Supplementary Fig. 3f). Similar results were obtained in BRCA1-knockout RPE1 cells (Fig. 3b). Moreover, BRCA1 or BRCA2 knockdown caused gap accumulation in wildtype HeLa cells, but not in PARP14-knockout HeLa cells we previously generated[67] (Fig. 3c). These findings indicate that PARP14 is essential for replication-associated ssDNA gap accumulation in BRCA-deficient cells.

Recently, inhibition of the MRE11 nuclease was shown to suppress gap accumulation in BRCA-deficient cells, as measured using the S1 nuclease fiber assay[35], suggesting that gap formation upon BRCA deficiency involves the nuclease activity of MRE11, similar to fork degradation in these cells. In accordance with this previous study, MRE11 depletion or inhibition also suppressed gap formation in RPE1-BRCA1[KO] and HeLa-BRCA2[KO] cells, as measured using the BrdU alkaline comet assay (Fig. 3b, d, Supplementary Fig. 3f). Co-depletion of PARP14 and MRE11 appeared to show an epistatic effect in RPE1 cells (Fig. 3b), but an additive one in HeLa cells (Supplementary Fig. 3f), potentially suggesting cell line differences in the case of this particular assay. Since we found that PARP14 mediates both replication fork degradation, and replication-associated gap formation, two processes which involve MRE11 activity, we sought to assess the impact of PARP14 on MRE11 recruitment to nascent DNA upon replication stress. We employed the SIRF assay which was previously utilized to measure MRE11 engagement on stalled replication forks in BRCA-deficient cells[30]. In line with this previous study, BRCA2-knockout HeLa and DLD1 cells showed increased MRE11 binding to nascent DNA upon HU treatment; However, depletion of PARP14 completely suppressed this increased MRE11 engagement (Fig. 3e–g; Supplementary Fig. 4a, b). Moreover, PARP14 depletion also suppressed the binding of MRE11 to nascent DNA in BRCA-proficient (wildtype) HeLa cells treated with the RAD51 inhibitor B02 (Fig. 3h). The reduction in MRE11 binding to nascent DNA upon PARP14 depletion was not caused by a decrease in overall MRE11 protein levels, since western blot experiments showed that PARP14 knockdown does not affect MRE11 levels (Fig. 3i). Overall, these findings indicate that PARP14 is essential for MRE11 recruitment to stalled replication forks.

## PARP14 interacts with MRE11 and binds nascent DNA
To understand how PARP14 participates in MRE11 recruitment to nascent DNA, we next investigated if PARP14 may directly interact with MRE11. A previously-published unbiased large-scale identification of

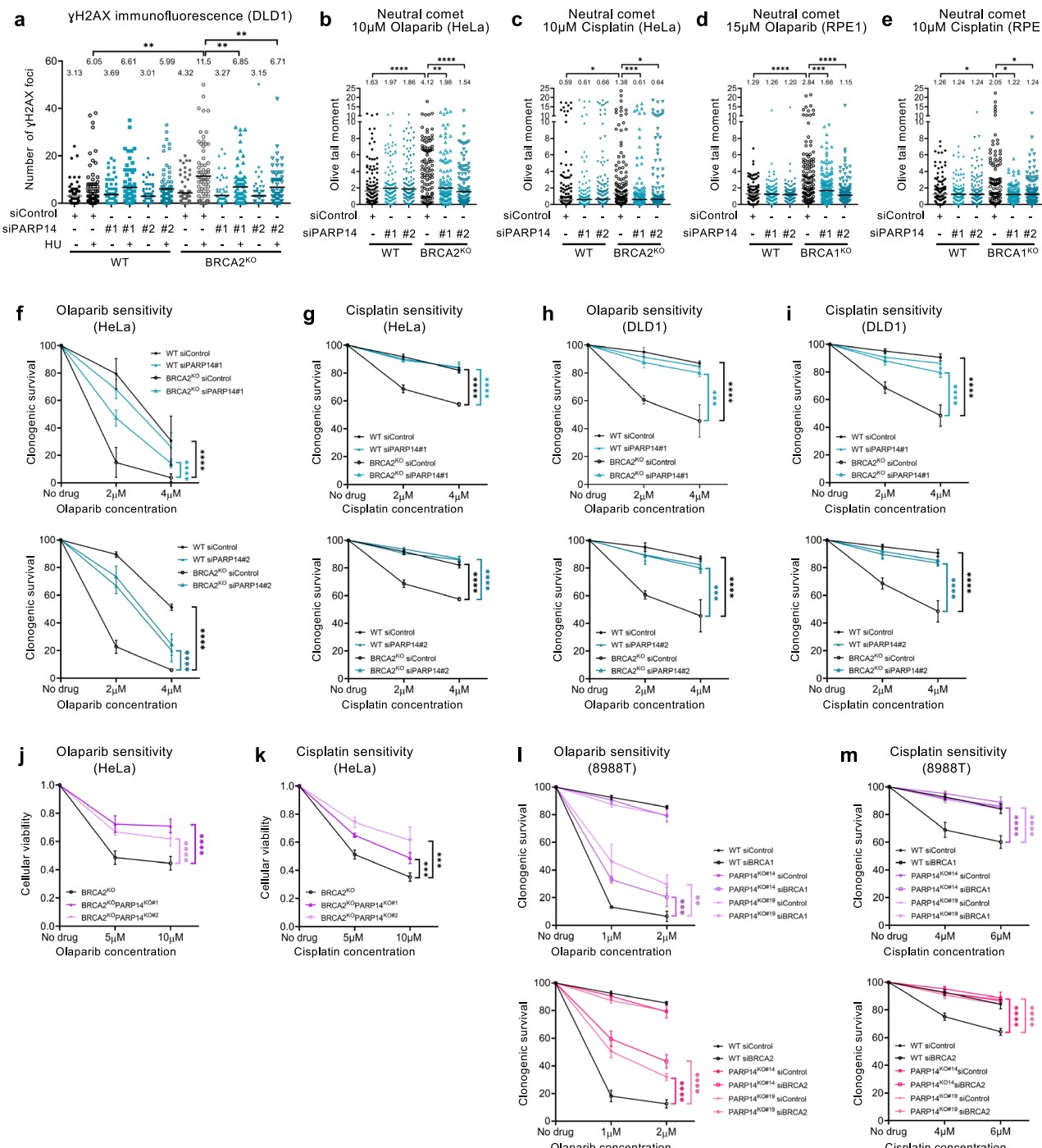

**Fig. 2 | Loss of PARP14 promotes chemoresistance in BRCA-deficient cells.**
**a** γH2AX immunofluorescence showing that PARP14 depletion reduces γH2AX foci upon HU treatment (4 mM HU for 4 h) in DLD1-BRCA2KO cells. At least 75 cells were quantified for each condition. The mean value is represented on the graphs, and asterisks indicate statistical significance (t-test two-tailed, unpaired). **b–e** Neutral comet assays showing that PARP14 depletion suppresses the accumulation of DNA damage induces by olaparib (**b**, **d**) or cisplatin (**c**, **e**) in HeLa-BRCA2KO (**b**, **c**) and RPE1-BRCA1KO (**d**, **e**) cells. At least 75 comets were quantified for each sample. The median values are marked on the graph and listed at the top. Asterisks indicate statistical significance (Mann-Whitney, two-tailed). **f–i** Clonogenic survival experiments showing that depletion of PARP14 increases the resistance of HeLa-BRCA2KO (**f**, **g**) and DLD1-BRCA2KO (**h**, **i**) cells to olaparib (**f**, **h**) and cisplatin (**g**, **i**) Two different siRNA oligonucleotides for PARP14 knockdown were used shown in different graphs). The average of 3 independent experiments (with the exception of the top

graph in **f**, which shows 4 independent experiments), with standard deviations indicated as error bars, is shown. Asterisks indicate statistical significance calculated using 2-way ANOVA. **j**, **k** Cellular viability assays showing that two independent HeLa-BRCA2KOPARP14KO double knockout cell lines have increased resistance to olaparib (**j**) and cisplatin (**k**) compared to HeLa-BRCA2KO cells. The average of four independent experiments, with standard deviations indicated as error bars, is shown. Asterisks indicate statistical significance calculated using 2-way ANOVA. **l**, **m** Clonogenic survival experiments showing that depletion of BRCA1 (top graphs) or BRCA2 (bottom graphs) in PARP14-knockout 8988 T cells causes resistance to olaparib (**l**) and cisplatin (**m**) compared to their depletion in wildtype cells. Two independent PARP14-knockout clones were used. The average of three independent experiments, with standard deviations indicated as error bars, is shown. Asterisks indicate statistical significance calculated using 2-way ANOVA. Source data are provided as a Source Data file.

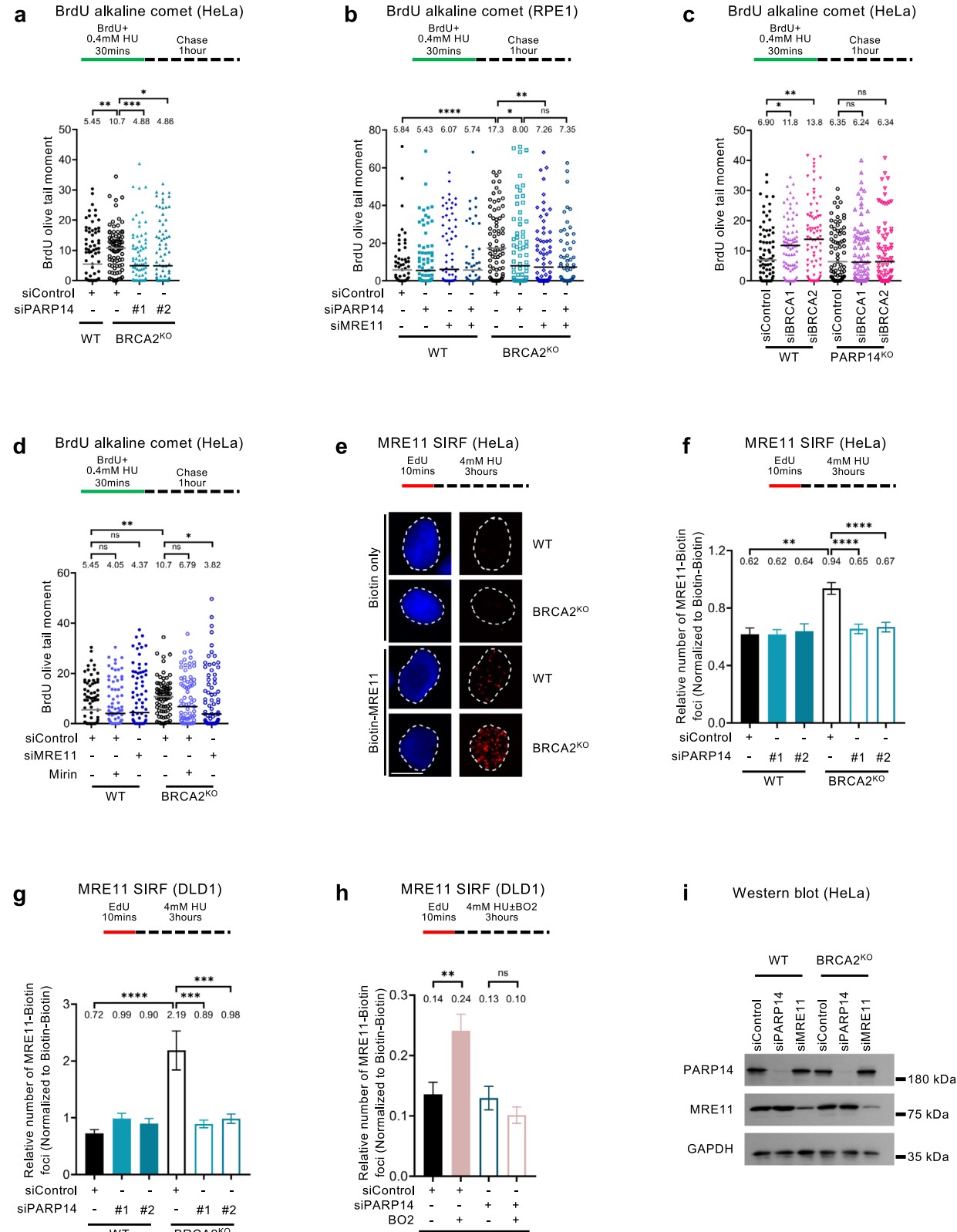

PARP14 interacting factors by BioID labeling coupled with mass spectrometry identified MRE11 as a potential PARP14 binding partner[15]. We employed the PLA assay to confirm these findings, and quantitatively measure the PARP14-MRE11 interaction under various physiological conditions. We could specifically detect MRE11-PARP14 PLA foci in wildtype cells, but not in PARP14-knockout HeLa cells (Fig. 4a; Supplementary Fig. 4c), thereby confirming this interaction. Moreover, an

interaction between MRE11 and PARP14 was also detected by co-immunoprecipitation experiments (Supplementary Fig. 4d). HU treatment resulted in an increase in PLA foci formation (Fig. 4b), indicating that replication stress promotes the PARP14-MRE interaction. In BRCA2-knockout cells, the PARP14-MRE11 PLA foci formation was identical to wildtype cells (Fig. 4b). These findings suggest that, even though the engagement of MRE11 on nascent DNA is enhanced in

**Fig. 3 | PARP14 promotes the engagement of the MRE11 nuclease on nascent DNA in BRCA-deficient cells. a, b** BrdU alkaline comet assay showing that PARP14 depletion reduces replication-associated ssDNA gaps accumulation upon HU treatment in HeLa-BRCA2[KO] (**a**) and RPE1-BRCA1[KO] (**b**) cells. At least 45 nuclei were quantified for each condition. The median values are marked on the graph and listed at the top. Asterisks indicate statistical significance (Mann-Whitney, two-tailed). A schematic representation of the assay conditions is shown at the top. **c** BrdU alkaline comet assay showing that depletion of BRCA1 or BRCA2 induces the accumulation of replication-dependent ssDNA gaps in wildtype, but not in PARP14-knockout HeLa cells. At least 75 nuclei were quantified for each condition. The median values are marked on the graph and listed at the top. Asterisks indicate statistical significance (Mann-Whitney, two-tailed). A schematic representation of the assay conditions is shown at the top. **d** BrdU alkaline comet assay showing that knockdown or inhibition of MRE11 reduces replication-associated ssDNA gaps accumulation upon HU treatment in HeLa-BRCA2[KO] cells. At least 75 nuclei were quantified for each condition. The median values are marked on the graph and listed at the top. Asterisks indicate statistical significance (Mann-Whitney,

two-tailed). A schematic representation of the assay conditions is shown at the top. **e–g** MRE11 SIRF experiment showing that PARP14 depletion reduces HU-induced MRE11 binding to nascent DNA in HeLa-BRCA2[KO] (**e, f**) and DLD1-BRCA2[KO] (**g**) cells. Representative micrographs, with scale bars representing 10 μm (**e**) and quantifications (**f, g**) are shown. At least 100 cells were quantified for each condition. Bars indicate the mean values, error bars represent standard error of the mean, and asterisks indicate statistical significance (t-test, two-tailed, unpaired). Schematic representations of the assay conditions are shown at the top. Single antibody controls are shown in Supplementary Fig. 4a. **h** MRE11 SIRF experiment showing that PARP14 depletion reduces HU-induced MRE11 binding to nascent DNA caused by treatment of wildtype HeLa cells with the RAD51 inhibitor B02. At least 100 cells were quantified for each condition. Bars indicate the mean values, error bars represent standard error of the mean, and asterisks indicate statistical significance (t-test, two-tailed, unpaired). A schematic representation of the assay conditions is shown at the top. **i** Western blot showing that PARP14 depletion does not affect MRE11 protein levels. Source data are provided as a Source Data file.

---

BRCA-deficient cells, the interaction between PARP14 and MRE11 occurs similarly in wildtype and BRCA-deficient cells.

Since our findings indicate that PARP14 interacts with MRE11 and promotes MRE11 engagement on nascent DNA, we next tested if PARP14 is itself recruited to nascent DNA. Indeed, PARP14 SIRF foci were specifically observed in wildtype cells, but not in PARP14-knockout HeLa cells (Fig. 4c; Supplementary Fig. 4e, f), indicating that PARP14 binds nascent DNA under normal growth conditions. PARP14 SIRF foci were increased in BRCA2-knockout HeLa and DLD1 cells compared to their wildtype counterparts, particularly upon HU treatment (Fig. 4d, e). This pattern is reminiscent of the binding of MRE11 to nascent DNA[30], and suggests that PARP14 and MRE11 may bind to stalled replication forks as a complex.

To further explore this possibility, we investigated the determinants of PARP14 recruitment to nascent DNA upon replication stress. HU-induced accumulation of PARP14 SIRF foci in BRCA2-knockout HeLa cells was suppressed by depletion of PARP1 and ZRANB3, which are involved in fork slowing and reversal, suggesting that, similar to MRE11, PARP14 binds reversed fork structures (Fig. 4f; Supplementary Fig. 4g). Interestingly, MRE11 knockdown also reduced PARP14 SIRF foci (Fig. 4f; Supplementary Fig. 4h, showing an interdependency between MRE11 and PARP14 for binding to stalled replication forks. Finally, PARP14 binding to nascent DNA was also induced by treatment of BRCA-proficient (wildtype) HeLa cells with the RAD51 inhibitor B02 (Supplementary Fig. 4i), which is known to cause MRE11-mediated fork degradation[30]. Overall, these findings suggest that PARP14 co-localizes with MRE11, in a mutually-dependent manner, on reversed forks undergoing resection.

## Impact of PARP14 catalytic activity in BRCA-deficient cells

PARP14 is a mono-ADP-ribosyltransferase with over one hundred putative targets identified through unbiased methods[15]. We thus sought to investigate if the PARP14 catalytic activity is important for the function of PARP14 in promoting nascent strand degradation in BRCA-deficient cells. Previously, mutation of the H1698 residue in mouse Parp14 protein was shown to abolish the catalytic activity[68]. Alignments of the catalytic domain sequences of the mouse and human PARP1 and PARP14 proteins identified H1682 as the corresponding site in human PARP14 (Supplementary Fig. 5a). To confirm that H1682 is essential for the catalytic activity of human PARP14, we recombinantly expressed in *E. coli* a fragment of the human PARP14 protein spanning the catalytic (PARP) domain (aminoacids 1470-1801/end) (Supplementary Fig. 5b), either wildtype or harboring the H1682Q mutation. We purified this fragment and performed in vitro ADP-ribosylation assays using biotinylated NAD + . Wildtype, but not H1682Q mutant PARP14 catalytic domain showed robust ADP-ribosylation activity in this assay (Supplementary Fig. 5c), indicating

that this mutant is devoid of catalytic activity. We next employed CRISPR/Cas9 to generate HeLa cells bearing homozygous H1682Q mutation in all PARP14 alleles (referred to as HeLa-PARP14[H1682Q] cells) (Supplementary Fig. 5d). Western blots showed that the mutant PARP14-H1682Q protein is expressed in these cells at lower levels than endogenous PARP14 in wildtype cells (Fig. 5a). However, we noticed that stable ectopic re-expression of wildtype PARP14 cDNA in HeLa-PARP14[KO] cells results in a relatively similar expression of PARP14 protein levels (Fig. 5a). Thus, we employed these cells as PARP14-wildtype control for experiments using the PARP14-H1682Q mutant.

First, we investigated if the PARP14-H1682Q mutant localizes to stalled forks similar to the wild-type form. In line with the results presented above in Fig. 4 with BRCA2-knockout cells, knockdown of BRCA1 or BRCA2 in control HeLa cells resulted in an increase in PARP14 SIRF foci upon HU treatment (Fig. 5b). In contrast, BRCA1 or BRCA2 depletion in HeLa-PARP14[H1682Q] cells did not affect PARP14 SIRF foci levels under these conditions (Fig. 5c). These findings suggest that the PARP14 catalytic activity is required for its localization to stalled replication forks.

Since the results presented above in Fig. 4 indicated a mutually-dependent binding of PARP14 and MRE11 to nascent DNA, we next tested if the defect in the recruitment of the PARP14-H1682Q mutant to stalled forks impacts MRE11-mediated degradation of nascent DNA in BRCA-deficient cells. We first measured MRE11 recruitment to stalled replication forks in the HeLa-PARP14[H1682Q] cells. In line with the experiments using BRCA2-knockout cells, depletion of BRCA1 or BRCA2 by siRNA resulted in HU-induced MRE11 SIRF foci in wild-type cells, but not in PARP14-knockout HeLa cells (Fig. 5d, e). Complementation of the PARP14-knockout cells by exogenous expression of wildtype PAR14 cDNA restored MRE11 recruitment to nascent DNA upon BRCA1 or BRCA2 depletion (Fig. 5f). However, BRCA1 or BRCA2 depletion in HeLa-PARP14[H1682Q] cells did not increase MRE11 SIRF foci (Fig. 5g), indicating that MRE11 recruitment to stalled forks is defective in PARP14 catalytic mutant cells.

We next investigated if this defective MRE11 localization in HeLa-PARP14[H1682Q] cells is associated with fork protection. In line with the results presented above in Fig. 1 with 8988 T cells, BRCA1 or BRCA2 depletion caused fork degradation in wildtype, but not in PARP14-knockout HeLa cells (Fig. 5h; Supplementary Fig. 5e). Complementation of the PARP14-knockout cells by exogenous expression of wildtype PARP14 cDNA restored fork degradation upon BRCA1 or BRCA2 depletion, indicating that, even though the exogenous protein is expressed at lower levels, it is able to functionally correct the knockout. In contrast, BRCA1 or BRCA2 depletion in HeLa-PARP14[H1682Q] cells did not cause fork degradation, similar to the situation in PARP14-knockout cells (Fig. 5h; Supplementary Fig. 5e). These findings indicate

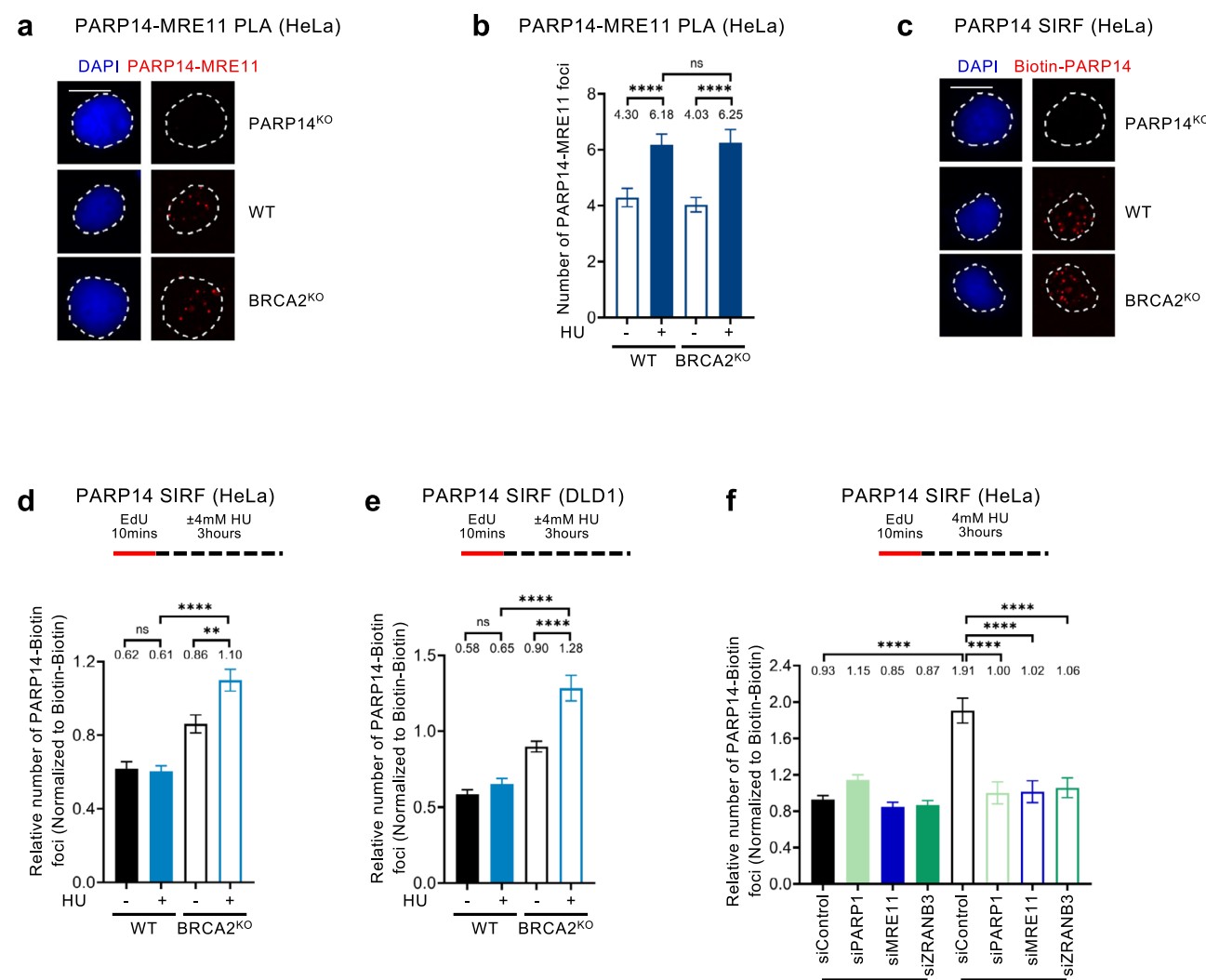

**Fig. 4 | PARP14 binds stalled replication forks in BRCA-deficient cells.**
**a**, **b** PARP14-MRE11 PLA experiments in HeLa cells showing that the interaction between PARP14 and MRE11 is increased by HU treatment, but is not affected by BRCA2 knockout. Representative micrographs, with scale bars representing 10 μm (**a**) and quantifications (**b**) are shown. The specificity of the readout is demonstrated by the loss of PARP14-MRE11 PLA foci in HeLa-PARP14^KO cells (**a**). At least 100 cells were quantified for each condition. Bars indicate the mean values, error bars represent standard error of the mean, and asterisks indicate statistical significance (t-test, two-tailed, unpaired). **c** Representative micrographs (with scale bars representing 10 μm) of PARP14 SIRF experiments demonstrating the specificity of the readout, since the PARP14 SIRF signal is not present in HeLa-PARP14^KO cells. Single antibody controls are shown in Supplementary Fig. 4e. **d**, **e** PARP14 SIRF

experiments showing that PARP14 binds to nascent DNA in BRCA2-knockout HeLa (**d**) and DLD1 (**e**) cells upon HU treatment. At least 100 cells were quantified for each condition. Bars indicate the mean values, error bars represent standard error of the mean, and asterisks indicate statistical significance (t-test, two-tailed, unpaired). Schematic representations of the assay conditions are shown at the top. **f** PARP14 SIRF experiment showing that depletion of PARP1, MRE11, or ZRANB3 suppresses HU-induced engagement of PARP14 on nascent DNA in HeLa-BRCA2^KO cells. At least 100 cells were quantified for each condition. Bars indicate the mean values, error bars represent standard error of the mean, and asterisks indicate statistical significance (t-test, two-tailed, unpaired). A schematic representation of the assay conditions is shown at the top. Western blots confirming the knockdowns are shown in Supplementary Fig. 4g, h. Source data are provided as a Source Data file.

that PARP14 catalytic activity is required for the role of PARP14 in mediating fork degradation in BRCA-deficient cells.

In line with this, depletion of BRCA1 or BRCA2 resulted in γH2AX foci accumulation upon HU treatment in PARP14-wildtype control HeLa cells, but not in PARP14-knockout cells or in HeLa-PARP14^HI682Q cells (Fig. 5i). Similarly, depletion of BRCA1 or BRCA2 caused HU-induced ssDNA gap accumulation in control cells, but not in HeLa-PARP14^HI682Q cells (Fig. 5j). Finally, BRCA1 or BRCA2 knockdown caused olaparib sensitivity in control cells, but not in HeLa-PARP14^KO cells or in HeLa-PARP14^HI682Q cells (Fig. 5k, l). These findings suggest that the PARP14 catalytic mutation suppresses MRE11-mediated genomic instability and renders BRCA-deficient cells chemoresistant.

To further establish this, we sought to complement the genetic approach presented above with a pharmacological approach. Specific

PARP14 inhibitors, which act by binding to regions of the catalytic site which are unique to PARP14 among all PARPs, have been recently developed[16,69] and are commercially available. Treatment of BRCA2-knockout HeLa cells with two different small molecule inhibitors of PARP14, namely H10 and RBN012759, suppressed fork degradation in BRCA2-knockout HeLa or DLD1 cells, as well as in BRCA1-knockout RPE1 cells, similar to the genetic loss of PARP14 (Fig. 6a–c). In line with this, MRE11 SIRF experiments indicated that the PARP14 inhibitors suppress MRE11 engagement on nascent DNA in HU-treated BRCA2-knockout cells (Fig. 6d). Moreover, similar to the results described above with the PARP14-H1682Q catalytic mutant, treatment of BRCA2-knockout HeLa cells with the PARP14 inhibitors suppressed HU-induced ssDNA gap accumulation in these cells (Fig. 6e). The inhibitors also suppressed PARP14 SIRF foci formation induced by BRCA1 or

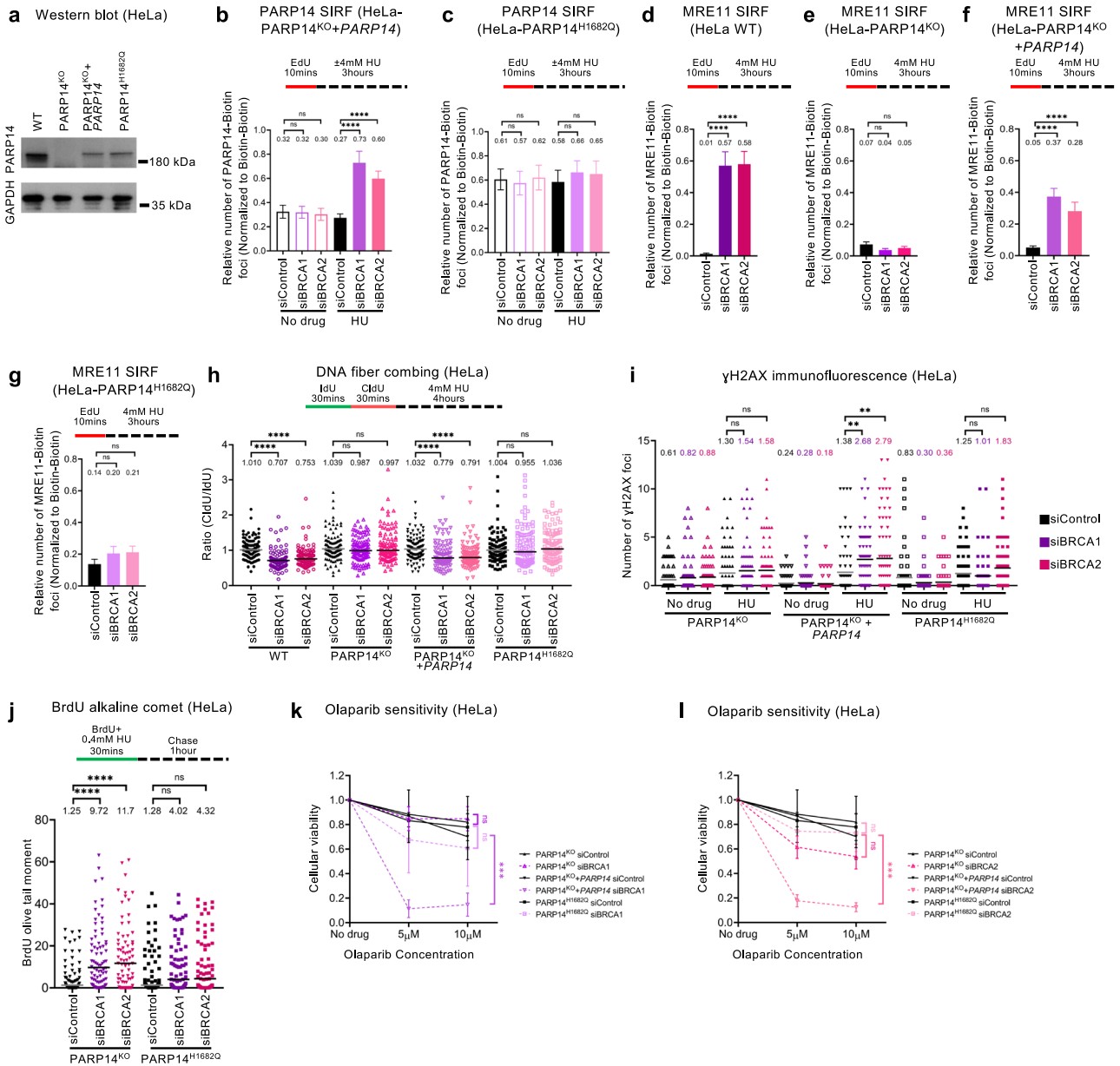

**Fig. 5 | The catalytic activity of PARP14 promotes genomic instability in BRCA-deficient cells. a** Western blot showing similar PARP14 levels in HeLa-PARP14[KO] cells complemented by stable re-expression of PARP14 cDNA, and in PARP14[H1682Q] mutant cells. **b, c** PARP14 SIRF experiments showing that knockdown of BRCA1 or BRCA2 increases PARP14 binding to nascent DNA upon HU treatment in HeLa-PARP14[KO] cells complemented by stable re-expression of PARP14 cDNA (**b**), but not in PARP14[H1682Q] mutant cells (**c**). At least 100 cells were quantified for each condition. Bars indicate the mean values, error bars represent standard error of the mean, and asterisks indicate statistical significance (t-test, two-tailed, unpaired). Schematic representations of the assay conditions are shown at the top. **d–g** MRE11 SIRF experiments showing that knockdown of BRCA1 or BRCA2 causes MRE11 engagement on nascent DNA upon HU treatment in wildtype HeLa cells (**d**) as well as in HeLa-PARP14[KO] cells complemented by stable re-expression of PARP14 cDNA (**f**), but not in HeLa-PARP14[KO] cells (**e**) or in PARP14[H1682Q] mutant cells (**g**). At least 100 cells were quantified for each condition. Bars indicate the mean values, error bars represent standard error of the mean, and asterisks indicate statistical significance (t-test, two-tailed, unpaired). Schematic representations of the assay conditions are shown at the top. **h** DNA fiber combing assay showing that knockdown of BRCA1 or BRCA2 causes HU-induced fork degradation in wildtype HeLa cells and in HeLa-PARP14[KO] cells complemented by stable re-expression of PARP14 cDNA, but not in HeLa-PARP14[KO] cells or in PARP14[H1682Q] mutant cells. The ratio of CldU to IdU tract lengths is presented, with the median values marked on the

graphs and listed at the top. At least 100 tracts were quantified for each sample. Asterisks indicate statistical significance (Mann-Whitney, two-tailed). A schematic representation of the assay conditions is shown at the top. Western blots confirming BRCA1 and BRCA2 depletion are shown in Supplementary Fig. 5e. **i** γH2AX immunofluorescence showing that knockdown of BRCA1 or BRCA2 causes HU-induced γH2AX foci formation in HeLa-PARP14[KO] cells complemented by stable re-expression of PARP14 cDNA, but not in HeLa-PARP14[KO] cells or in PARP14[H1682Q] mutant cells. At least 75 cells were quantified for each condition. The mean value is represented on the graphs, and asterisks indicate statistical significance (t-test two-tailed, unpaired). **j** BrdU alkaline comet assay showing that knockdown of BRCA1 or BRCA2 causes accumulation of replication-associated ssDNA gaps upon HU treatment in HeLa-PARP14[KO] cells complemented by stable re-expression of PARP14 cDNA, but not in PARP14[H1682Q] cells. At least 75 nuclei were quantified for each condition. The median values are marked on the graph and listed at the top. Asterisks indicate statistical significance (Mann-Whitney, two-tailed). A schematic representation of the assay conditions is shown at the top. **k, l** Cellular viability assays showing that knockdown of BRCA1 (**k**) or BRCA2 (**l**) causes olaparib sensitivity in HeLa-PARP14[KO] cells complemented by stable re-expression of PARP14 cDNA, but not in HeLa-PARP14[KO] cells or in PARP14[H1682Q] mutant cells. The average of three independent experiments, with standard deviations indicated as error bars, is shown. Asterisks indicate statistical significance calculated using 2-way ANOVA. Source data are provided as a Source Data file.

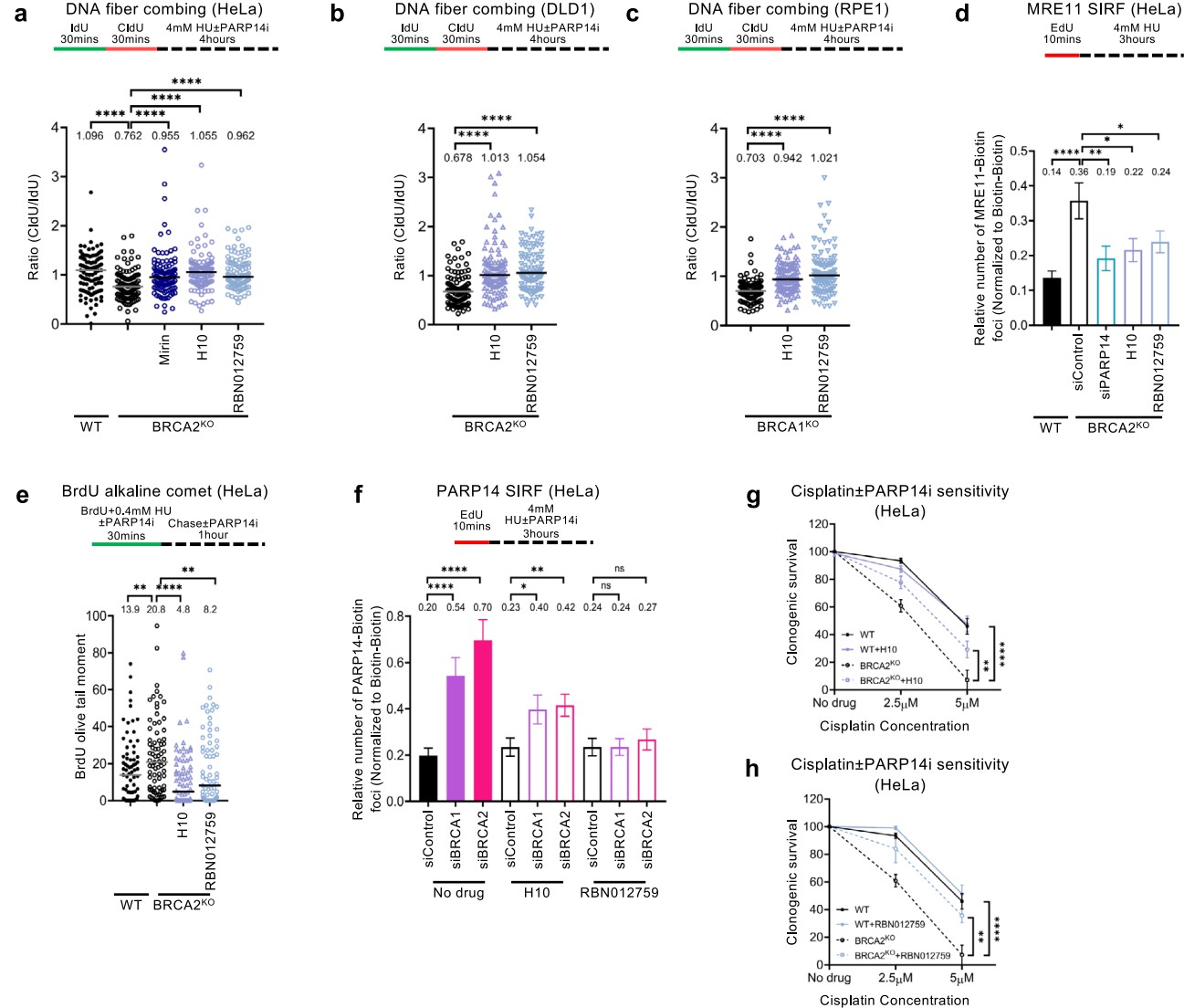

**Fig. 6 | Inhibition of PARP14 suppresses MRE11-mediated nucleolytic degradation of nascent DNA in BRCA-deficient cells. a–c** DNA fiber combing assay showing that treatment with two separate PARP14 inhibitors, namely H10 and RBN012759 suppresses HU-induced fork degradation in HeLa-BRCA2^KO (**a**), DLD1-BRCA2^KO (**b**) and RPE1-BRCA2^KO (**c**) cells, similar to treatment with the MRE11 inhibitor mirin. The ratio of CldU to IdU tract lengths is presented, with the median values marked on the graphs and listed at the top. At least 100 tracts were quantified for each sample. Asterisks indicate statistical significance (Mann-Whitney, two-tailed). A schematic representation of the assay conditions is shown at the top. **d** MRE11 SIRF experiment showing that treatment with PARP14 inhibitors H10 and RBN012759 suppresses HU-induced MRE11 binding to nascent DNA in HeLa-BRCA2^KO cells. At least 100 cells were quantified for each condition. Bars indicate the mean values, error bars represent standard error of the mean, and asterisks indicate statistical significance (t-test, two-tailed, unpaired). A schematic representation of the assay conditions is shown at the top. **e** BrdU alkaline comet assay showing that treatment with PARP14 inhibitors H10 and RBN012759 suppresses

HU-induced accumulation of replication-associated ssDNA gaps in HeLa-BRCA2^KO cells. At least 75 nuclei were quantified for each condition. The median values are marked on the graph and listed at the top. Asterisks indicate statistical significance (Mann-Whitney, two-tailed). A schematic representation of the assay conditions is shown at the top. **f** PARP14 SIRF experiment showing that treatment with PARP14 inhibitors H10 and RBN012759 suppresses PARP14 binding to nascent DNA in BRCA1 or BRCA2-depleted HeLa cells. At least 100 cells were quantified for each condition. Bars indicate the mean values, error bars represent standard error of the mean, and asterisks indicate statistical significance (t-test, two-tailed, unpaired). A schematic representation of the assay conditions is shown at the top. **g, h** Clonogenic survival experiments showing that treatment with PARP14 inhibitors H10 (10μM) (**g**), or RBN012759 (10μM) (**h**) promotes resistance to cisplatin in HeLa-BRCA2^KO cells. The average of 3 independent experiments, with standard deviations indicated as error bars, is shown. Asterisks indicate statistical significance calculated using 2-way ANOVA. Source data are provided as a Source Data file.

BRCA2 depletion (Fig. 6f), confirming the findings with the H1682Q mutant described above, and thus further indicating that the PARP14 catalytic activity is required for its recruitment to stalled replication forks. Finally, these inhibitors, while mildly enhancing cisplatin sensitivity in wildtype cells, suppressed the cisplatin hypersensitivity of BRCA2-knockout HeLa cells (Fig. 6g, h; Supplementary Fig. 5f). Altogether, these findings unambiguously show that the catalytic activity of PARP14 is required for the genomic instability and chemosensitivity of BRCA-deficient cells.

## KU differentially regulates fork resection by MRE11 and EXO1
The KU complex binds DSB ends[41–43]. Interestingly, members of the KU complex were previously identified as putative PARP14 interacting partners in an unbiased proximity labeling-based pulldown[15]. To validate this, we performed PLA assays and could specifically detect PARP14-KU80 PLA foci in wildtype, but not in PARP14-knockout HeLa cells (Fig. 7a; Supplementary Fig. 6a), thus validating the PARP14-KU colocalization. Interestingly, the PARP14-KU80 PLA foci were increased in BRCA2-knockout cells and particularly upon HU

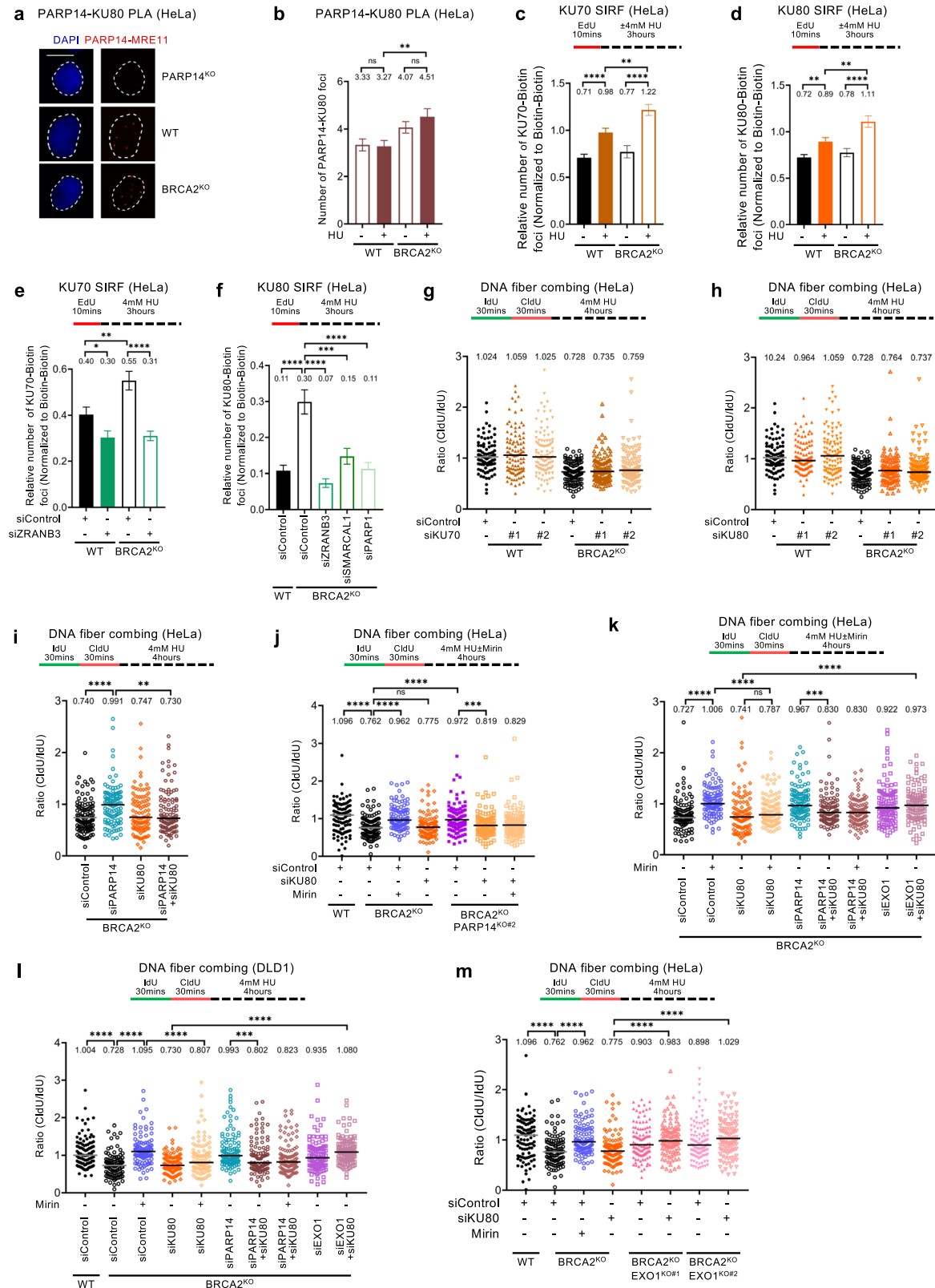

treatment (Fig. 7b), similar to the pattern we described above for PARP14 binding to nascent DNA (Fig. 4).

We next sought to investigate if the KU complex itself binds stalled replication forks. SIRF experiments indicated that both KU70 and KU80 were recruited to nascent DNA, and this was enhanced in BRCA2-knockout HeLa cells and upon HU treatment (Fig. 7c, d; Supplementary Fig. 6b–d). Importantly, this increase was suppressed by

depletion of ZRANB3, SMARCAL1 or PARP1 (Fig. 7e, f; Supplementary Fig. 6e), suggesting that KU binds to the exposed DSB end of reversed replication forks in BRCA-deficient cells.

We next investigated if KU impacts the degradation of stalled replication forks in human cells. DNA fiber combing assays indicated that KU70 or KU80 depletion does not induce nascent strand degradation in HU-treated wildtype cells (Fig. 7g, h; Supplementary Fig. 6f),

**Fig. 7 | Differential regulation of nascent strand degradation by EXO1 and MRE11 by the KU complex. a, b** PARP14-KU80 PLA experiments in HeLa cells showing that the interaction between PARP14 and KU is increased by BRCA2 deficiency. Representative micrographs, with scale bars representing 10 μm (**a**) and quantifications (**b**) are shown. The specificity of the readout is demonstrated by the loss of PARP14-KU80 PLA foci in HeLa-PARP14$^{KO}$ cells (**a**). At least 100 cells were quantified for each condition. Bars indicate the mean values, error bars represent standard error of the mean, and asterisks indicate statistical significance (t-test, two-tailed, unpaired). **c, d** KU SIRF experiments showing that KU70 (**c**) and KU80 (**d**) binding to nascent DNA is increased by HU treatment and BRCA2 deficiency. At least 100 cells were quantified for each condition. Bars indicate the mean values, error bars represent standard error of the mean, and asterisks indicate statistical significance (t-test, two-tailed, unpaired). Schematic representations of the assay conditions are shown at the top. Validation of the KU SIRF readout is shown in Supplementary Fig. 6b–d. **e, f** KU SIRF experiments showing that KU70 (**e**) and KU80 (**f**) binding to nascent DNA upon HU treatment in HeLa-BRCA2$^{KO}$ cells is suppressed by knockdown of fork reversal factors ZRANB3, SMARCAL1 and PARP1. At least 100 cells were quantified for each condition. Bars indicate the mean values, error bars represent standard error of the mean, and asterisks indicate statistical significance (t-test, two-tailed, unpaired). Schematic representations of the assay conditions are shown at the top. Western blots confirming the depletions are shown in Supplementary Fig. 6e. **g, h** DNA fiber combing assays showing that knockdown of KU70 (**g**) or KU80 (**h**), with two different siRNA oligonucleotides for

each, does not impact HU-induced fork degradation in HeLa-BRCA2$^{KO}$ cells. Western blots confirming KU70 and KU80 depletion are shown in Supplementary Fig. 6f. **i** DNA fiber combing assays showing that KU80 co-depletion restores fork degradation in PARP14-depleted HeLa-BRCA2$^{KO}$ cells. Western blots showing co-depletion of PARP14 and KU80 are presented in Supplementary Fig. 6g. **j** DNA fiber combing assays showing that KU80 depletion restores fork degradation in HeLa-BRCA2$^{KO}$PARP14$^{KO}$ double knockout cells, and this degradation is not performed by MRE11 since it is not rescued by mirin treatment. **k, l** DNA fiber combing assays showing that KU80 co-depletion restores fork degradation in PARP14-depleted BRCA2-knockout cells, and this degradation is not performed by MRE11 since it is not rescued by mirin treatment. Co-depletion of EXO1, but not inhibition of MRE11 by mirin, restores fork protection in KU80-depleted BRCA2-knockout cells. Similar results were obtained in HeLa-BRCA2$^{KO}$ (**k**) and DLD1-BRCA2$^{KO}$ (**l**) cells. Western blots showing co-depletion of EXO1 and KU80 are presented in Supplementary Fig. 6h. **m** DNA fiber combing assays showing that EXO1 knockout restores fork protection in KU80-depleted HeLa-BRCA2$^{KO}$ cells. Two independent BRCA2$^{KO}$EXO1$^{KO}$ double knockout clones were analyzed. Western blots confirming the EXO1 and BRCA2 knockout are shown in Supplementary Fig. 6i. For panels **g−m**, the ratio of CldU to IdU tract lengths is presented, with the median values marked on the graphs and listed at the top. At least 100 tracts were quantified for each sample. Asterisks indicate statistical significance (Mann-Whitney, two-tailed). Schematic representations of the assay conditions are shown at the top. Source data are provided as a Source Data file.

indicating that KU is not essential for fork protection in BRCA-proficient cells. Moreover, KU70 or KU80 depletion did not affect the HU-induced fork degradation observed in BRCA2-knockout cells (Fig. 7g, h). However, we surprisingly noticed that co-depletion of KU80 abolished fork protection conferred by loss of PARP14, by knockdown or knockout, in BRCA2-deficient HeLa cells (Fig. 7i, j; Supplementary Fig. 6g), suggesting that KU is required for the activity of PARP14 in promoting fork degradation in BRCA-deficient cells. In addition, mirin treatment, while suppressing fork degradation in BRCA2-knockout HeLa or DLD1 cells, was unable to do so upon KU80 depletion in these cells (Fig. 7j–l). This indicates that KU is required for MRE11-mediated fork degradation.

The inability of mirin treatment to rescue fork degradation in KU-deficient BRCA-mutant cells also implies that fork degradation in the absence of KU is not dependent on MRE11, but instead on a different nuclease. Since KU was shown to suppress EXO1-mediated DSB resection, we sought to test if EXO1 is the nuclease responsible for fork degradation in KU-deficient BRCA-mutant cells. Indeed, unlike mirin treatment, co-depletion of EXO1 was able to restore fork protection in KU80-depleted BRCA2-knockout cells (Fig. 7k, l; Supplementary Fig. 6h), indicating that EXO1 is the nuclease responsible for fork degradation in these cells. To further validate these findings without the need to perform co-depletion studies, we employed CRISPR/Cas9 to knock-out EXO1 in HeLa-BRCA2$^{KO}$ cells (Supplementary Fig. 6i). Similar to the co-depletion studies, knockout of EXO1 caused fork protection in KU80-depleted HeLa-BRCA2$^{KO}$ cells (Fig. 7m), confirming that, in KU-deficient cells, EXO1 but not MRE11 performs fork degradation. Overall, these findings suggest that KU binding to reversed forks promotes PARP14-MRE11-dependent degradation of stalled replication forks in BRCA-deficient cells, but suppresses EXO1-mediated degradation.

To gain insights into this regulation, we employed the SIRF assay to measure the impact of KU on the recruitment of PARP14, MRE11, and EXO1 to nascent DNA. Strikingly, KU70 or KU80 depletion abolished the recruitment of both PARP14 and MRE11 to nascent DNA in HU-treated BRCA2-knockout HeLa and DLD1 cells (Fig. 8a–d) and BRCA2$^{KO}$EXO1$^{KO}$ double knockout HeLa cells (Supplementary Fig. 6j, k), indicating that KU binding to the DSB-like end of the reversed fork is essential for recruitment of the MRE11-PARP14 complex, and explaining why MRE11 is not mediating the fork degradation in KU-deficient BRCA-mutant cells.

We next investigated the recruitment of EXO1 to nascent DNA. SIRF experiments showed that, unlike MRE11 recruitment, EXO1 is not

recruited to stalled forks in BRCA2-knockout DLD1 cells; however, KU depletion resulted in significant increase in EXO1 SIRF foci in these cells, which was suppressed by ZRANB3 co-depletion (Fig. 8e; Supplementary Fig. 6l). These findings indicate that KU suppresses EXO1 engagement on reversed replication forks.

If KU suppresses EXO1 engagement, why is EXO1 required for fork degradation in KU-proficient BRCA2-mutant cells[27] (Fig. 7k–m)? We reasoned that EXO1 engagement on nascent DNA in KU-proficient cells takes place at a later stage in fork degradation, after MRE11 mediates the removal of KU80. To address this, we increased the EdU labeling time in the SIRF assay from 10 min (which was the standard labeling time used in the previous experiments) to 30 min, to capture EXO1 binding to nascent DNA further away from the DNA end. Under these conditions, we observed a large increase in EXO1 SIRF foci in BRCA2-knockout HeLa and DLD1 cells, even though KU was still present in these cells; This increase was dependent on PARP14 and MRE11 (Fig. 8f, g). Both endonuclease and exonuclease activities of MRE11 were required for EXO1 SIRF foci formation under these conditions (Fig. 8h) -in line with the findings reported above that both the endonuclease and the exonuclease activities of MRE11 are required for fork degradation in these cells (Supplementary Fig. 1a−c). In contrast, inhibition of MRE11 or of PARP14 increased KU SIRF foci formation (Supplementary Fig. 6m). Overall, these findings indicate that, in BRCA-deficient cells, the PARP14-MRE11 complex is recruited to KU bound on reversed replication forks, to initiate short-patch fork degradation through sequential engagements of the endo- and subsequently exonuclease activities of MRE11; this results in KU removal, allowing EXO1 engagement and excessive EXO1-mediated long-patch fork resection (Fig. 8i).

## Discussion

Genomic instability is a hallmark of BRCA-deficient cells, which underlies BRCA-associated carcinogenesis. Moreover, restoration of genome protection is associated with chemotherapeutic resistance of BRCA-deficient tumors[22,23,30−33]. Several mechanisms of genomic stability are known to be mediated by the BRCA pathway, including: HR-mediated DSB repair, replication fork protection, and more recently, suppression of ssDNA gap accumulation. In BRCA-deficient cells, both fork degradation and ssDNA gap accumulation occur through the unabated activity of the MRE11 nuclease[29,35]. Thus, in-depth understanding of MRE11 regulation is important. It is generally considered that the fork protection activity of BRCA2 involves loading of RAD51 on

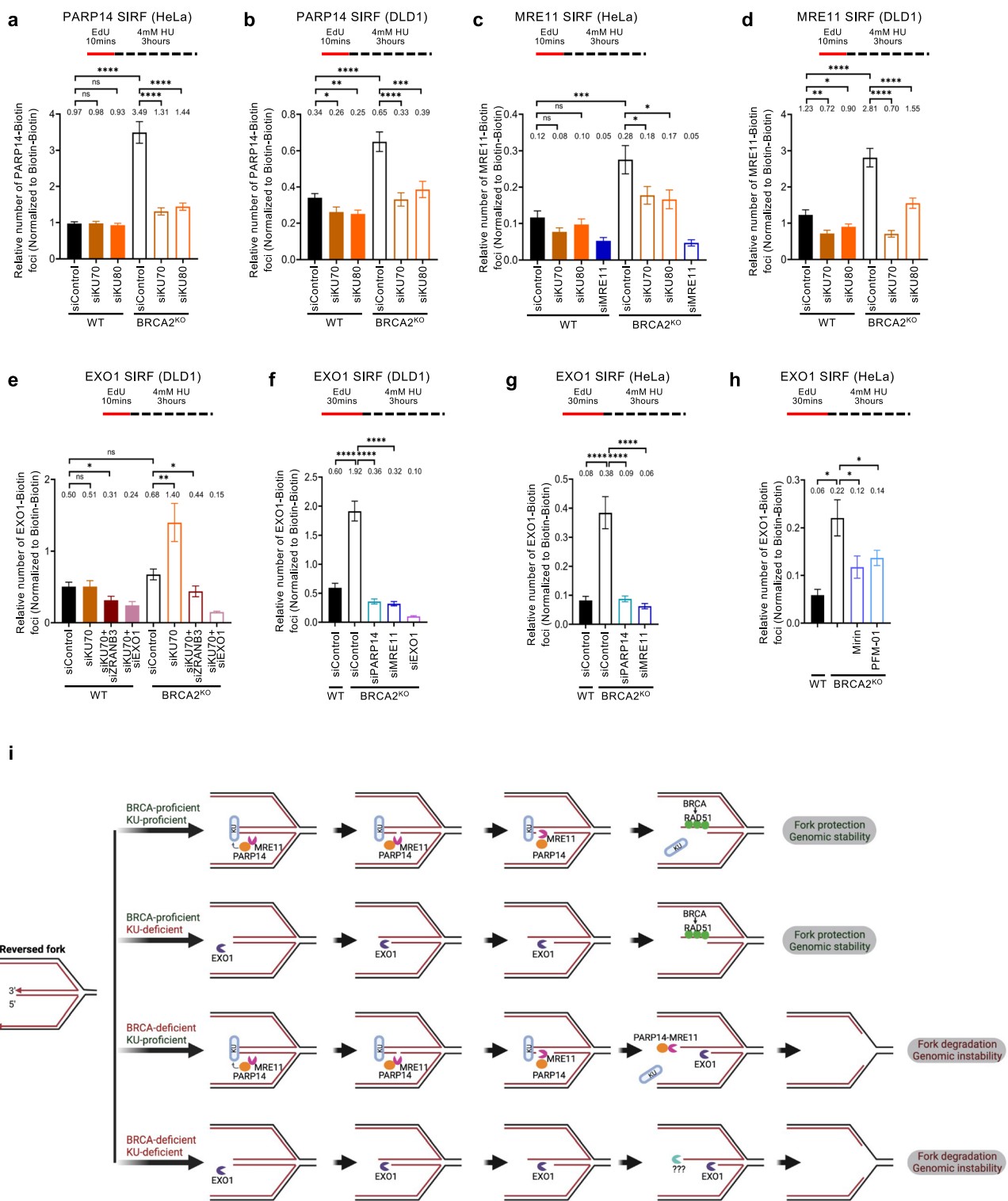

reversed forks, which keeps MRE11 in check, thus suppressing excessive resection[26,29]. Whether RAD51 loading is also a factor in BRCA-mediated suppression of ssDNA gap accumulation is less clear[35]. Here, we identify PARP14 as an MRE11-binding factor which modulates its engagement on nascent DNA. We show that PARP14 itself binds nascent DNA upon replication stress, and PARP14-deficient cells have defective MRE11 recruitment to stalled replication forks, suggesting that PARP14 promotes MRE11 recruitment to DNA. Importantly, PARP14 is required for both MRE11 activities which cause genome instability in BRCA-deficient cells, namely fork degradation and ssDNA gap formation. While we cannot rule out an impact of PARP14 loss on

recombination restoration in BRCA-deficient cells, these findings potentially explain why loss of PARP14 suppresses genomic instability in BRCA-deficient cells, and renders them chemoresistant. Our work suggests that PARP14 levels may impact the response to BRCA-deficient tumors to genotoxic chemotherapy, with potential clinical implications.

While nascent strand degradation was shown to occur on reversed forks[21–23,28,30], replication-dependent gap formation and/or expansion by MRE11 is suggested to take place behind the replication fork, such as upon fork re-priming by the PRIMPOL primase[35]. We speculate that PARP14 mediates recruitment of MRE11 to both reversed forks, and the

**Fig. 8 | KU promotes MRE11 binding to nascent DNA in BRCA-deficient cells, but suppresses binding of EXO1. a–d** SIRF experiments showing that binding of PARP14 (**a**, **b**) and MRE11 (**c**, **d**) to nascent DNA in BRCA2-knockout HeLa (**a**, **c**) and DLD1 (**b**, **d**) cells is suppressed by depletion of KU70 or KU80. **e** EXO1 SIRF experiment showing that EXO1 is not recruited to nascent DNA in HeLa-BRCA2[KO] cells upon HU treatment. However, depletion of KU70 results in binding of EXO1 to nascent DNA under these conditions. ZRANB3 co-depletion suppresses this binding. EXO1 co-depletion reduces the signal, showing the specificity of the EXO1 SIRF readout. Single antibody controls are shown in Supplementary Fig. 6l. **f–h** EXO1 SIRF experiments showing binding of EXO1 to nascent DNA in DLD1-BRCA2[KO] (**f**) and HeLa-BRCA2[KO] (**g**, **h**) when the EdU labeling time was increased from 10 mins to 30 mins. EXO1 co-depletion reduces the signal, showing the specificity of the EXO1 SIRF readout. For all panels, at least 100 cells were quantified for each condition. Bars indicate the mean values, error bars represent standard error of the mean, and asterisks indicate statistical significance (t-test, two-tailed, unpaired). Schematic representations of the assay conditions are shown at the top. **i** Schematic representation of the proposed model. KU binds the exposed DSB end of symmetrical reversed forks, protecting it against EXO1. At the same time, KU bound on the reversed fork recruits the PARP14-MRE11 complex, and through its endonuclease activity MRE11 creates a nick, which is then process by its 3′ to 5′ exonuclease activity towards the DSB end. This results in release of KU from the DSB end. In BRCA-proficient cells, loading of RAD51 stabilizes the ssDNA overhang against further nucleolytic processing. In the absence of KU, EXO1 engages the DSB end with its 5′ to 3′ exonuclease activity for, but loading of RAD51 by the BRCA pathway on the partially resected DSB end stabilizes it against further degradation. In BRCA-deficient cells, the partially resected structure is susceptible to continuous (long-range) degradation by EXO1 on the 5′ end strand, and MRE11 (and potentially other 3′ to 5′ exonucleases) on the 3′ end strand. Created with BioRender.com. Source data are provided as a Source Data file.

initial gaps or nicks which are then expanded by MRE11 to form ssDNA gaps. What is the mechanistic similarity between those two structures that makes PARP14 a required factor in both events is unclear.

Using both pharmacological PARP14 inhibition and genetic PARP14 catalytic mutant inactivation, we show that PARP14 catalytic activity is required for promoting MRE11-mediated genomic instability in BRCA-deficient cells. Compared to other post-translational modifications, understanding the functional impact of ADP-ribosylation, and in particular of MARylation, on particular substrates has been a much more difficult task[1]. MARylation is difficult to detect in cells, and is less specific in terms of substrates and substrate residues modified, which makes it difficult to specifically inactivate it for functional studies. While specific readers of this modification have been described, it is unclear to what extent MARylation works by recruiting readers to the modified substrate, or signals functional changes through different means. Moreover, unbiased proteomic substrate identification approaches showed that many PARP enzymes, including PARP14, have hundreds of potential targets, spread out across diverse biological pathways. One of this potential PARP14 substrates is RAD50[15], a member of the MRE11-NBS1-RAD50 (MRN) complex, raising the possibility that RAD50 MARylation by PARP14 may be involved in MRE11 recruitment to nascent DNA.

Another PARP14 interactor and potential substrate previously identified through unbiased proximity ligation-based methods[15] is the KU complex. Here, we show that KU binds reversed replication forks in human cells, where it differentially regulates the engagement of nucleases: KU promotes MRE11-mediated fork degradation, while suppressing EXO1-mediated fork degradation. Since the recruitment of both PARP14 and MRE11 to nascent DNA upon replication stress is suppressed upon loss of KU, we speculate that KU bound to the DSB end at the reversed fork, recruits the PARP14-MRE11 complex to engage in fork degradation (Fig. 8i). Indeed, previously-reported studies have suggested that KU is required for MRE11 recruitment to DNA damage foci[70]. An interaction between KU and MRE11 has been previously reported using co-immunoprecipitation[70] and proximity ligation[71] assays. Moreover, super-resolution fluorescent microscopy indicated co-localization of KU and MRE11 on the same DNA molecule at/near a DSB end[71,72]. Finally, work in DSB repair model systems indicated that KU directs the MRE11 endonuclease activity to initiate resection, which then results in KU eviction[50,51].

Since PARP14 interacts with both MRE11 and KU, and is required for MRE11 recruitment, we moreover speculate that, in cells, PARP14 may bridge the MRE11 interaction with KU (Fig. 8i). Alternatively, PARP14 may independently be recruited to nascent DNA through its RRM motif, a motif which was previously shown to be able to interact with ssDNA[73].

Our work indicates that processing of reversed forks occurs in a manner similar to DNA end resection, with MRE11 endonuclease activity, perhaps with CTIP as a cofactor, initiating a nick which is then extended outward towards the DSB end by the 3′ to 5′ exonuclease activity of MRE11, and inward towards the fork junction by the 5′ to 3′ exonuclease activity of EXO1 (Fig. 8i). In our model, KU binding to the reversed end is essential for PARP14-mediated recruitment of MRE11. In turn, MRE11 engagement processes the end in a manner incompatible with KU binding, resulting in its release. In KU-deficient cells, recruitment of the PARP14-MRE11 complex is impaired, but the unprotected DSB end is attacked by EXO1. In wildtype cells, through RAD51 loading, or perhaps though direct inhibition of nucleases, BRCA proteins stabilize the partly-resected reversed end by suppressing excessive resection by MRE11 and/or EXO1. In BRCA-deficient cells, long-range resection by EXO1 is ultimately the cause of the fork degradation observed.

Our model (Fig. 8i) applies to reversed fork which are symmetrical, forming a blunt-ended DSB which is the substrate for KU binding. Such reversed forks necessitate controlled resection, in order to expose ssDNA stretches for RAD51 loading by BRCA2[26]. In contrast, an asymmetrical reversed fork would not be bound by KU, but in this case the exposed ssDNA would be sufficient for RAD51 loading by BRCA2, as proposed in the initial model of BRCA2-mediated fork protection[29]. If also PARP14 plays a role in the engagement of MRE11 on asymmetrical reversed forks remains to be determined.

## Methods

### Cell culture and protein techniques

HeLa, RPE1, 8988 T, U2OS, and MDA-MB-436 cells were grown in Dulbecco's modified Eagle's media (DMEM). DLD-1 cells were grown in Roswell Park memorial Institute (RPMI) 1640 media. Media was supplemented with 15% FBS and penicillin/streptomycin. 8988T-PARP14[KO] cells[20], HeLa-PARP14[KO] cells[67], and HeLa-BRCA2[KO] cells[56] were generated in our laboratory and previously described. DLD1-BRCA2[KO] cells (Horizon HD105-007) were obtained from Dr. Robert Brosh (National Institute on Aging, Baltimore, MD). RPE1-BRCA1[KO] (also harboring p53 homozygous deletion) were obtained from Dr. Alan D'Andrea (Dana-Farber Cancer Institute, Boston, MA)[74]. U2OS DR-GFP cells were obtained from Dr. Jeremy Stark (City of Hope National Medical Center, Duarte, CA)[65]. MDA-MB-436 cells were obtained from Dr. Hong-Gang Wang (Penn State College of Medicine). To re-express exogenous PARP14 in the knockout cell lines, cells were infected with the lentiviral construct pLV-Puro-SV40 > Flag/hPARP14 (VectorBuilder), constitutively expressing Flag-tagged PARP14 under the control of the SV40 promoter. To knock-out PARP14 or EXO1 in HeLa-BRCA2[KO] cells, the commercially available CRISPR/Cas9 KO plasmids for PARP14 (Santa Cruz Biotechnology sc-402812) and EXO1 (Santa Cruz Biotechnology sc-402356) were used. Transfected cells were FACS-sorted into 96-well plates using a BD FACSAria II instrument. Resulting colonies were screened by Western blot. The HeLa-PARP14[H1682Q] cell line was created using CRISPR/Cas9-mediated genome editing by Vector-Builder, using the gRNA sequence: CTCTTCCATGGGACAGATGC, and

the ssODN repair template: GCAAAGAAAAAAACTATGGATGCCAA GAATGGCCAGACAATGAATGAGAAGCAACTCTTCCAGGGCACTGACG CCGGCTCCGTGCCACACGTCAATCGAAATGGCTTTAACCGCAGCTA TGCCGGAAAGAATGG. Individual clones were analyzed by sequencing of the genomic region amplified by PCR (forward primer: TTCATG-CACCGGTCTTCCAA; reverse primer: CTAGAAGGGCCAGTCAATCCC).

Gene knockdown was performed using Lipofectamine RNAiMAX. AllStars Negative Control siRNA (Qiagen 1027281) was used as control. The following oligonucleotide sequences (Stealth or SilencerSelect siRNA, ThermoFisher) were used:

PARP14#1: AGGCCGACUGUGACCAGAUAGUGAA;
PARP14#2: CGGCACUACACAGUGAACUUGAACA;
PARP14#3: UAGCACAGAAGAUUCUUGCACUUUA;
PARP10: GCCUGGUGGAGAUGGUGCUAUUGAU;
PARP1: AAACAUGGGCGACUGCACCAUGAUG;
ZRANB3: UGGCAAUGUAGUCUCUGCACCUAUA;
BRCA1: AAUGAGUCCAGUUUCGUUGCCUCUG;
BRCA2: GAGAGGCCUGUAAAGACCUUGAAUU;
MRE11: AGAAACAUGUUGGUUUGCUGCGUAU;
EXO1: CCUGUUGAGUCAGUAUUCUCUUUCA;
SMARCAL1: CACCCUUUGCUAACCCAACUCAUAA;
KU70#1: Assay ID s52594;
KU70#2: Assay ID 10066;
KU80#1: Assay ID s14952;
KU80#2: Assay ID 139858.

Denatured whole cell extracts were prepared by boiling cells in 100 mM Tris, 4% SDS, 0.5 M β-mercaptoethanol. Antibodies used for Western blot, at 1:500 dilution, were:

PARP14 (Santa Cruz Biotechnology sc-377150);
PARP10 (Novus NB100-2157);
BRCA1 (Santa Cruz Biotechnology sc-6954);
BRCA2 (Calbiochem OP95);
ZRANB3 (Invitrogen PA5-65143);
PARP1 (Cell Signaling Technology 9542 S);
MRE11 (Santa Cruz Biotechnology sc-135992);
KU70 (Abcam ab83501);
KU80 (Abcam ab119935);
EXO1 (Novus NBP2-16391);
SMARCAL1 (Invitrogen PA5-54181);
GAPDH (Santa Cruz Biotechnology sc-47724).

Specific inhibitors used were: Olaparib (Selleck Chemicals S1060), mirin (Selleck Chemicals S8096), PFM01 (Tocris 6222), B02 (Millipore Sigma SML0364); H10 (Tocris 6228); RBN012759 (Medchemexpress HY-136979).

## Drug sensitivity assays

For clonogenic survival assays, 500 siRNA-treated cells were seeded per well in 6-well plates and incubated with the indicated doses of olaparib or cisplatin. Media was changed after 3 days and colonies were stained after 10–14 days. Colonies were washed with PBS, fixed with a solution of 10% methanol and 10% acetic acid, and stained with 2% crystal violet (Aqua solutions). To assess cellular viability, a luminescent ATP-based assay was performed using the CellTiterGlo reagent (Promega G7572) according to the manufacturer's instructions. Following treatment with siRNA, 1500 cells were seeded per well in 96-well plated and incubated with the indicated drug doses of olaparib or cisplatin for 3 days. Apoptosis assays were performed using the FITC Annexin V kit (Biolegend, 640906). Quantification was performed on a BD FACSCanto 10 flow cytometer using the FlowJo v10 software.

## Functional cellular assays

For the DR-GFP homologous recombination assay[65], GFP-positive cells were detected by flow cytometry 3 days after I-SceI transfection. Neutral and BrdU alkaline comet assays were performed[40] using the

Comet Assay Kit (Trevigen, 4250-050). For the BrdU alkaline comet assay, cells were incubated with 100 μM BrdU as indicated. Drugs (4 mM HU, 50 μM mirin, 20 μM H10, or 25 μM RBN012759) were added according to the labeling schemes presented. Slides were imaged on a Nikon microscope operating the NIS Elements V1.10.00 software. Olive tail moment was analyzed using CometScore 2.0. Immunofluorescence was performed[75] using a γH2AX antibody (Millipore Sigma JBW301).

## DNA fiber assays

Cells were incubated with 100 μM IdU and 100 μM CldU as indicated. Drugs (4 mM or 0.4 mM HU, 25 μM B02, 50 μM mirin, 100 μM PFM01, 20 μM H10, or 25 μM RBN012759) were added according to the labeling schemes presented. Next, cells were collected and processed using the the FiberPrep kit (Genomic Vision EXT-001) according to the manufacturer's instructions. DNA molecules were stretched onto coverslips (Genomic Vision COV-002-RUO) using the FiberComb Molecular Combing instrument (Genomic Vision MCS-001). Slides were then stained with antibodies detecting CldU (Abcam 6236), IdU (BD 347580), and DNA (Millipore Sigma MAD3034) and incubated with secondary Cy3 (Abcam 6946), Cy5 (Abcam 6565), or BV480 (BD Biosciences 564879) conjugated antibodies. Finally, the cells were mounted onto coverslips and imaged using a confocal microscope (Leica SP5) and analyzed using LASX 3.5.7.23225 software.

## Proximity ligation-based assays

For PLA assays, cells were seeded into 8-chamber slides and 24 h later, were treated with 4 mm HU for 3hrs as indicated. Cells were then permeabilized with 0.5% Triton for 10 min at 4 C, washed with PBS, fixed at room temperature with 3% paraformaldehyde in PBS for 10 min, washed again in PBS and then blocked in Duolink blocking solution (Millipore Sigma DUO82007) for 1 h at 37 C, and incubated overnight at 4 C with primary antibodies. The primary antibodies used were: PARP14 (Santa Cruz Biotechnology sc-377150); MRE11 (Genetex GTX70212); KU70 (Santa Cruz Biotechnology sc-56092); KU80 (Abcam ab-119935). Samples were then subjected to a proximity ligation reaction using the Duolink kit (Millipore Sigma DUO92008) according to the manufacturer's instructions. Slides were imaged using a Deltavision microscope with SoftWorx 6.5.2 software, and images were analyzed using ImageJ 1.53a software. At least 100 cells were quantified for each sample.

For SIRF assays, cells were seeded into 8-chamber slides and 24 h later they were pulse-labeled with 50 μM EdU for 10 min or 30 min, followed by drug treatment (4 mM HU, 25 μM B02, 20 μM H10, or 25 μM RBN01275) for 3 h as indicated. Cells were permeabilized with 0.5% Triton for 10 min at 4 C, washed with PBS, fixed at room temperature with 3% paraformaldehyde in PBS for 10 min, washed again in PBS, and then blocked in 3% BSA in PBS for 30 min. Cells were then subjected to Click-iT reaction with biotin-azide using the Click-iT Cell Reaction Buffer Kit (ThermoFisher, C10269) for 30 min and incubated overnight at 4 C with primary antibodies diluted in PBS with 1% BSA. The primary antibodies used were: Biotin (mouse: Jackson ImmunoResearch 200-002-211; rabbit: Bethyl Laboratories A150-109A); PARP14 (Santa Cruz Biotechnology sc-377150); MRE11 (GeneTex GTX70212); EXO1 (Santa Cruz Biotechnology sc-56092); KU70 (Santa Cruz Biotechnology sc-5309); KU80 (Abcam ab-119935), PARP1 (Cell Signaling Technology 9542). Next, samples were subjected to a proximity ligation reaction using the Duolink kit (Millipore Sigma DUO92008) according to the manufacturer's instructions. Slides were imaged using a Deltavision microscope and images were analyzed using ImageJ 1.52p software. At least 100 cells were quantified for each sample. To account for variation in EdU uptake between samples, for each sample, the number of protein-biotin foci were normalized to the average

number of biotin-biotin foci for that respective sample. The scale bars for the PLA and SIRF micrographs shown represent 10 μm.

## Statistics and reproducibility

For clonogenic and cellular viability assays, the 2-way ANOVA statistical test was used when multiple concentrations are shown in line graphs. For bar graphs (drug sensitivity assays where only one concentration is shown, Annexin V assays, DR-GFP assays), the t-test (two-tailed, unpaired) was used. For both line and bar graphs, the results shown are from independent biological replicates. For immunofluorescence and proximity ligation assays, the t-test (two-tailed, unpaired) was used. For the DNA fiber assay and the comet assay, the Mann-Whitney statistical test was performed. For immunofluorescence, DNA fiber combing, proximity ligation, and comet assays, results from one experiment are shown; the results were reproduced in at least one additional independent biological conceptual replicate. Western blot experiments were reproduced at least two times. Statistical analyses were performed using GraphPad Prism 9 and Microsoft Excel v2205 software. Statistical significance is indicated for each graph (ns = not significant, for $p > 0.05$; * for $p \leq 0.05$; ** for $p \leq 0.01$; *** for $p \leq 0.001$, **** for $p \leq 0.0001$).

## Reporting summary

Further information on research design is available in the Nature Research Reporting Summary linked to this article.

## Data availability

The data generated during this study are available from the corresponding author upon reasonable request. Source data are provided with this paper.

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

## Acknowledgements

We would like to thank Drs. Jeremy Stark, Robert Brosh, Hong-Gang Wang, Michael O'Connor and Alan D'Andrea for materials, as well as Tanay Thakar, Joshua Straka, Anastasia Hale, and Aurosman Pappus Sahu for advice and support; and the following Penn State College of Medicine core facilities: Flow Cytometry, Genomic Analyses, and Imaging. Experimental design schemes and models were created with Biorender.com. This work was supported by: NIH R01ES026184 and NIH R01GM134681 (to G.L.M.), and NIH R01CA244417 (to C.M.N.).

## Author contributions

A.D., C.M.N. and G.L.M. designed and conducted the experiments, and wrote the paper.

## Competing interests

The authors declare no competing interests.
