## [Peer Review File · Nature Communications]

The KU-PARP14 axis differentially regulates DNA resection at stalled replication forks by MRE11 and EXO1REVIEWER COMMENTS

Reviewer #1 (Remarks to the Author):

In this manuscript, Dhoonmoon and colleagues describe a novel role for the mono-ADP ribosylation factor PARP14 in mediating the nucleolytic degradation of nascent DNA at remodelled, stalled replication forks in BRCA-deficient cells. The mechanisms described are novel, and represent a significant advance in our understanding of how DNA at these sites is processed. The central tenet of this work is that the catalytic function of PARP14 at reversed forks is required to recruit MRE11, which then drives nascent DNA degradation in BRCA-deficient cells in concert with EXO1, as well as the formation of ssDNA gaps (the suppression of which is a more recently-described function of BRCA1/2). The authors also suggest that the Ku70/80 heterodimer functions to protect DNA ends against EXO1-mediated DNA degradation, but also functions to recruit the PARP14-MRE11 module. This is all consistent with the theory that MRE11 functions at the regressed DNA arm of reversed forks in much the same way that it does at breaks – namely, to recognise the Ku-blocked DNA end to initiate an endonucleolytic incision that subsequently acts as an entry point for 3-5' exonucleolytic digestion by MRE11. In the process, Ku is removed, facilitating continued resection. Evidence in support of these hypotheses is provided largely in the form of DNA fibre assays, COMET assays, markers of DNA damage, survival assays (with Olaparib and Cisplatin), and SIRF assays.

Overall, this is a solid manuscript based on sound experimental techniques that have been correctly employed. I find myself in the rare position of having little to criticise. Among the main strengths of the manuscript is the depth of validation – the authors have gone to significant lengths to validate their findings. Experiments have been performed in multiple cell lines with various combinations of PARP14, BRCA1, BRCA2, Ku70, and Ku80 deficiency. siRNA-mediated depletion experiments are (often reciprocally) complemented throughout with CRISPR deletion studies, suggesting that the chances of any of the observed phenotypes being due to off-target effects are slim to none. This is further mitigated by phenotypic complementation experiments.

DNA fibre assays measuring IdU:CldU ratio have been employed throughout to assess degradation of nascent DNA at stalled forks. This is a standard assay and the data appears robust, and appropriately analysed. As controls, MRE11 involvement has been triaged using the established inhibitor mirin, and the role of stalled fork remodelling by depletion of the DNA translocase ZRANB3.

COMET assays and gammaH2AX foci formation assays have been carried out to assess the impact of the genetic manipulations performed on genome stability.

SIRF assays were performed to assess the interactions between PARP14, MRE11, Ku and nascent DNA. There are a few inconsistencies in this data in terms of the severity of some of the SIRF phenotypes reported – as an example of this, I refer to Figures 5N and 7C – which both measure the MRE11:biotin signal in WT vs BRCA2 KO HeLa cells. In one instance, the signal jumps by 3-fold in the BRCA KO cells, but in the other, the signal increases by barely 20% (though this is still statistically significant). However, I would mention that we have performed similar assays in our laboratory and have consistently observed an increase in MRE11 interaction with nascent DNA by PLA assays in BRCA2 KO cells. I am therefore confident that the trend is correct – but I suggest that the experiment in Fig 7C is repeated to confirm the result.

The question remains as to what precisely the activity of PARP14 is in the context of reversed forks/ssDNA gaps – an issue that the authors allude to in the discussion section. This remains unanswered, but I think that the data provided is already mechanistic enough to be of interest to the readership of Nature Comms. Furthermore, the authors suggest (very probably correctly) that the mutation of H1682 in human PARP14 is equivalent to H1698 in murine PARP14, which was previously shown to be catalytically inactive. Nonetheless, it would complement the manuscript if the authors could attempt some validation of defective catalysis. The commercially available Af1521 macrodomain beads are excellent at pulling down both MARylated and PARylated proteins. The authors could fairly

easily carry out analysis of an established PARP14 target such as STAT1 or PARP9 in macrodomain bead-mediated pulldowns from WT, PARP14 KO, PARP14 KO+ WT PARP14 and PARP14 KO+H1682 PARP14 cells, to validate that the mutant is in fact catalytically dead. In fact, these samples would be a very useful resource which the authors could employ to probe for RAD50 and other candidates that they consider potential PARP14 MARYlation targets following treatment with genotoxins (though for me these latter studies extend beyond the scope of this manuscript).

I might also suggest that in a few places, inclusion of representative images of DNA fibres, COMET tails and gammaH2AX foci might be a useful addition that would provide visual cues for the less expert reader.

The M+Ms section contains sufficient information for the data to be reproduced.

I have suggested two sets of experiments that I hope will improve the manuscript. However, overall I consider this work suitable for publication in Nature Communications, and expect it to be of significant interest to readers with an interest in genome stability and/or cancer cell biology.

Reviewer #2 (Remarks to the Author):

Replication fork reversal, which involves the annealing of the nascent strands of the sister chromatids, is a process whereby replication forks that have arrested due to replication stress may be processed for restart. Protection of reversed replication fork from MRE11 nuclease-driven nascent DNA degradation by the BRCA pathway is crucial to maintain genomic stability and chemoresistance. In this context, the manuscript by Dhoonmoon et al. sheds new light on the factors that are involved in this process, as well as the underlying mechanisms.

Key novelties include the identification of a novel role for the mono-ADP-ribosyltransferase PARP14 in orchestrating the engagement of the MRE11 nuclease on reversed replication forks in BRCA-deficient cells. Notably, this function requires the catalytic activity of PARP14. Specifically, loss of PARP14 suppressed MRE11-mediated fork degradation and gap accumulation, and promoted genome stability and chemoresistance in BRCA-deficient cells. In addition, the authors identify a physical interaction between PARP14 and the DNA end-binding complex KU complex and provide evidence that KU operates as a regulator of PARP14-MRE11-mediated DNA resection. They show that in human cells, KU binds the reversed fork end, as previously shown in *Sc. Pombe* by the Lambert's group. In addition, their data indicate that KU protects reversed replication forks from degradation mediated by EXO1. The authors integrate PARP14 as a new actor in a model that is reminiscent of the processing of single-ended DSBs resulting from the collapse of replication forks upon collision e.g. with ssDNA. In this model, KU eviction from the DNA end generated at reversed forks is promoted by PARP14-mediated MRE11 end/exo-nuclease activities and allows long-range resection by EXO1.

This is a very elegant and original investigation that uses a broad range of cellular models generated by RNAi and CRISPR-Cas9, as well as small-molecule inhibitors, in very complementary, high-profile experiments addressing the presence and interaction of DNA repair factors at nascent DNA.

Although hundreds of PARP14 MARYlation substrates have been identified, the substrates of PARP14 in this story remain unknown. However, as the authors pointed out correctly, MARYlation modifications are hard to study. Thus, the current lack of knowledge on the targets of PARP14 during reversed fork processing should not prevent this manuscript from being considered as a significant improvement in our understanding of the mechanisms underlying reversed fork processing.

However, I would like to raise the following issues that, in my opinion, should be addressed before this manuscript is accepted.

My major concerns relate to the last experiments carried out by the authors in order to support the

model presented in Figure 7H.

- Re: Fig.7F,7G

To support the claim that KU eviction (associated with PARP14-mediated MRE11 activity) allows EXO1 engagement and long patch fork resection, it would be important to examine KU presence by SIRF after 30 min of EdU labelling, in order to reveal a correlation between the presence of EXO1 observed at this time and loss of KU.

In addition, similar to what has been done for seDSBs, and to complement their experiments with a siMRE11, the authors should show that in cells where MRE11 exonuclease activity is inhibited (or better, in RNAI-mediated MRE11-depleted cells expressing an ectopic, endo/exo-defective MRE11 construct), KU is retained significantly while EXO1 is not recruited.

- Re: Fig 7A-D.

From the data presented in Fig 7A-D, and the observation that KU and PARP14 interact - as assessed by PLA (Fig 6A) - the authors suggests that KU is responsible for PARP14 recruitment at reversed replication forks. However, the authors should rule out that the lack of PARP14/MRE11 recruitment observed upon loss of KU in BRCA2-deficient cells is not due to the activity of EXO1, which could prevent such recruitment (e.g. due to nucleolytic degradation) and/or displace PARP14 from the DNA. In this context, it would be important to examine whether PARP14 can be detected at reversed replication forks when BRCA2-, KU-defective cells also express a catalytically-dead EXO1 construct. This is important because the authors have previously reported data indicating an interaction between PARP14 and PCNA (Ref 19), raising the possibility that other recruitment routes to (stalled) replication forks may exist for PARP14.

Other points:

- INTRODUCTION (last sentence of page 3): "Specific inhibitors of PARP14 catalytic activity have recently been developed, and targeting PARP14 has been proposed as a possible therapeutic approach for multiple cancer types (10, 13, 14, 16-18). "

Please specify some of the reasons why PARP14 is seen as a relevant therapeutic target.

RESULTS

- Fig. 1D: The authors should examine the impact of co-depleting ZRANB3 and PARP14, as this may reveal additional roles for PARP14 upstream from fork degradation. In addition, in this experiment, cells with siRNA-PARP14 behave like siCTL cells. How consistent is this with previous findings (Ref 19) by the authors regarding the hypersensitivity of PARP14-depleted cells to HU and the accumulation of gH2AX foci in these cells?

- Fig. 2A: What concentration of HU was used in this experiment with DLD1 cells? How long were the cells exposed to HU?

- Fig. 2F: In their previous ref 19, the authors showed that exposure of HeLa cells to 0.05 microM Olaparib resulted in a clonogenic survivals of about 40% (siCTL) and 20% (siPARP14-1). Here, the clonogenic survival are much higher, even at much higher doses of Olaparib. Why is this?

- Fig. S3A-B: Can the authors report the statistical significance of other relevant comparisons? Indeed, it looks like the impact of depleting PARP14, BRCA1 and BRCA2 is similar in these cells.

- Fig. 3C: I would recommend showing images of the biotin-MRE11 foci without DAPI counterstaining but with nuclear contours instead, to facilitate their visualization. This should also apply to other figures showing DAPI-counterstained nuclei (e.g. Fig 4A, 4C, 6A).

- Replace "exogenous" by "ectopic" in this sentence: "we noticed that stable exogenous re-expression of wildtype".

- Fig. 5B-G: the spacing in the titles of these figures should be changed to increase readability.

- In the following sentence the authors should also provide the underlying mechanism of the PARP14 inhibitors: "Specific PARP14 inhibitors have been recently developed (16, 64) and are commercially available."

- Fig. 5O. In this figure, for both inhibitors, it appears that the viability of cells exposed to the inhibitor or the combination [inhibitor + cisplatin] is almost identical in WT and mutant cells. The only difference here is the initial sensitivity of WT and BRCA2 KO cells to cisplatin. The impact of the inhibitors alone on cell viability is identical in WT and mutant cells. As this assay was carried out using a single concentration of cisplatin, and in order to control that the results do not reflect limitations of the ATP-dependent cell viability assay, I would recommend to re-do this experiment using clonogenicity assays over multiple cisplatin concentration.

- Fig. 7C: Can the authors comment on the MRE11 SIF data in HeLa cells versus DLD1 cells. MRE11 binding to nascent DNA seems to be less impacted in HeLa compared to DLD1 cells. Does this suggest other recruitment mechanisms in these cells? The authors should use a siMRE11 control in 7C, similar to the siExo control of 7E.

- Add "nascent" in the following sentence as such: We reasoned that EXO1 engagement on "nascent" DNA in KU-proficient cells takes place at a later stage in fork degradation.

Reviewer #3 (Remarks to the Author):

In this manuscript Dhoonmoon et al. assess the role of PARP14 in replication fork protection and processing. They propose that PARP14 targets Mre11 to stalled replication forks and as a consequence, loss of PARP14 suppresses Mre11-dependent degradation of replication forks in BRCA-deficient cells. These events are dependent on PARP14 catalytic activity, which is proposed to regulate the process by promoting accumulation of PARP14 at sites of replication stress. They also test the role of Ku70/80 in these events and propose that Ku targets PARP14/Mre11 to regressed forks, allowing partial resection that releases Ku to subsequently promote Exo1-dependent DNA resection.

The mechanism of processing replication forks and how this contributes to genome stability and DNA damage tolerance is topical. This work identifies an important role for PARP14 in this process, which is novel. The manuscript is well written and clearly presented. Generally the conclusion drawn are appropriate. As a whole, the experiments are well designed and executed. However, there are areas where additional controls would strengthen the study and these have been highlighted below.

Specific comments:

Data in figure 1A needs to be assessed using siRNA of Mre11 as opposed to Mirin. This will also offer the opportunity to assess whether Mre11 and PARP14 are epistatic with regards suppressing replication fork degradation in BRCA-deficient cells. This is a critical experiment to place Mre11 and PARP14 in the same pathway, as suggested here and throughout the manuscript. Similar to observations in Figs 1E and F, these experiments should be repeated in other cell lines.

Fig 1D: Fork slowing is only a surrogate marker of fork reversal and could be caused by a variety of factors. An alternative needs to be used to assess fork reversal (e.g. electron microscopy or recognition of forks by GFP-RuvA; www.nature.com/articles/s41467-019-09196-9).

Is restoration of replication fork degradation by expression of PARP14 dependent on the catalytic or DNA binding activities? Catalytic activity is tested later, but could a DNA binding domain mutant of PARP14 also be used to test if it is no longer able to restore replication fork degradation in BRCA2 cells? This might also help assess separation of function of PARP14 catalytic activity and DNA binding.

As a whole, the data in Fig 2 support the proposed hypothesis – that PARP14 disruption suppresses DNA damage and sensitivity of BRCA-deficient cells to cisplatin/olaparib. However, there are some experimental design issues and inconsistencies in data that need to be addressed. Fig 2 J&K: Parental cells and PARP14 KO cells need to be included in this analysis. Without these controls the experiment is difficult to interpret. Fig 2K: PARP14 KO#2 data, whilst statistically significant, is not dramatic. Fig 2L: The rescue of olaparib sensitivity with PARP14 KO is not dramatic. Untreated cells need to be included in Supp Fig, S2.

Using the DR-GFP assay to measure HR efficiency indicate that chemoresistance upon PARP14 loss is not due to restoration of HR. However, this assay is not measuring HR in the context of replication stress, but at a Isce1-induced DSB. It would be more informative to look at HR events associated with S-phase, such as sister chromatid exchanges, for example.

Fig 3: The suppression of Gap accumulation on PARP14 loss needs to be assessed in cell types other than HeLa. Given Mre11 also suppresses gap accumulation, it should be tested whether PARP14 and Mre11 are epistatic with regards this phenotype. This is critical to support the hypothesis that they function in the same pathway, as suggested in Fig 2D-H and subsequent figs in the manuscript.

PLA does not necessarily indicate 2 proteins interact, just that they get to similar regions of the genome. A co-IP needs to be performed to say the 2 proteins interact. Controls of Mre11 and PARP14 antibodies alone need to be included in Fig 4B. These controls (antibodies alone) should be included in all PLA throughout the manuscript.

Fig 4F: Parp1 has also been implicated in maintaining reversed forks by inhibiting RecQ, so the siPARP1 phenotype is not necessarily due to fork slowing.

Fig 5A-G: Is there a reason different cells are shown in different panels? Are these independent experiments? It would be more appropriate to compare cell types within the same experiments and this should be performed.

The PARP14 H1682Q mutant should be tested for a lack of catalytic activity in vitro. Additionally, a potential explanation for the data presented is that the H1682Q mutation impacts on the structure of PARP14 to disrupt DNA binding activity independently of catalytic activity. Do PARP14 inhibitors similarly prevent interaction of PARP14 with nascent DNA and/or PLA with Mre11? This is straightforward and should be tested. Can the DNA binding activity of PARP14 H1682Q be tested in vitro? Are there other mutations that can be exploited to separate the functions of catalytic inactivation and DNA binding? It is important that these possibilities are tested.

Parp14 KO cells should be included in Fig 5J.

The manuscript would be strengthened by inclusion of data testing the importance of PARP14 catalytic activity in promoting fork degradation in other cell lines. Given the availability of PARP14 inhibitors, this should be straightforward.

Fig 6: For PLA experiments, controls need to be included using PARP14 and Ku80 antibodies alone. The data comparing WT and PARP14 KO cells need to be quantified.

The suppression of Ku SIRF signal by depletion of ZRANB3 is not dramatic for Ku70. Can this be improved by depleting other fork regression factors (e.g. SMARCAL1, PARP1 etc.) alone, or in combinations?

Response to referees

We would like to thank the reviewers for their helpful and constructive comments, which led to a significantly improved manuscript. To address the reviewers' concerns, we are submitting a substantially revised manuscript with 40 new figure panels, as well as 2 revised figure panels and 2 figures for reviewers. Below, please find our point-by-point reply to reviewers' comments (our responses in red font).

Reviewer #1

We are very happy that the reviewer found that “*this is a solid manuscript based on sound experimental techniques that have been correctly employed*” and that “*the data appears robust, and appropriately analysed*”. We thank the reviewer for their helpful comments, which we have addressed as indicated below.

(Remarks to the Author):

In this manuscript, Dhoonmoon and colleagues describe a novel role for the mono-ADP ribosylation factor PARP14 in mediating the nucleolytic degradation of nascent DNA at remodelled, stalled replication forks in BRCA-deficient cells. The mechanisms described are novel, and represent a significant advance in our understanding of how DNA at these sites is processed. The central tenet of this work is that the catalytic function of PARP14 at reversed forks is required to recruit MRE11, which then drives nascent DNA degradation in BRCA-deficient cells in concert with EXO1, as well as the formation of ssDNA gaps (the suppression of which is a more recently-described function of BRCA1/2). The authors also suggest that the Ku70/80 heterodimer functions to protect DNA ends against EXO1-mediated DNA degradation, but also functions to recruit the PARP14-MRE11 module. This is all consistent with the theory that MRE11 functions at the regressed DNA arm of reversed forks in much the same way that it does at breaks – namely, to recognise the Ku-blocked DNA end to initiate an endonucleolytic incision that subsequently acts as an entry point for 3-5' exonucleolytic digestion by MRE11. In the process, Ku is removed, facilitating continued resection. Evidence in support of these hypotheses is provided largely in the form of DNA fibre assays, COMET assays, markers of DNA damage, survival assays (with Olaparib and Cisplatin), and SIF assays.

Overall, this is a solid manuscript based on sound experimental techniques that have been correctly employed. I find myself in the rare position of having little to criticise. Among the main strengths of the manuscript is the depth of validation – the authors have gone to significant lengths to validate their findings. Experiments have been performed in multiple cell lines with various combinations of PARP14, BRCA1, BRCA2, Ku70, and Ku80 deficiency. siRNA-mediated depletion experiments are (often reciprocally) complemented throughout with CRISPR deletion studies, suggesting that the chances of any of the observed phenotypes being due to off-target effects are slim to none. This is further mitigated by phenotypic complementation experiments.

DNA fibre assays measuring IdU:CldU ratio have been employed throughout to assess degradation of nascent DNA at stalled forks. This is a standard assay and the data appears robust, and appropriately analysed. As controls, MRE11 involvement has been triaged using the established inhibitor mirin, and the role of stalled fork remodelling by depletion of the DNA translocase ZRANB3.

COMET assays and gammaH2AX foci formation assays have been carried out to assess the impact of the genetic manipulations performed on genome stability.

SIRF assays were performed to assess the interactions between PARP14, MRE11, Ku and nascent DNA. There are a few inconsistencies in this data in terms of the severity of some of the SIRF phenotypes reported – as an example of this, I refer to Figures 5N and 7C – which both measure the MRE11:biotin signal in WT vs BRCA2 KO HeLa cells. In one instance, the signal jumps by 3-fold in the BRCA KO cells, but in the other, the signal increases by barely 20% (though this is still statistically significant). However, I would mention that we have performed similar assays in our laboratory and have consistently observed an increase in MRE11 interaction with nascent DNA by PLA assays in BRCA2 KO cells. I am therefore confident that the trend is correct – but I suggest that the experiment in Fig 7C is repeated to confirm the result.

We have repeated the experiment in the original Figure 7C and provide a new figure showing a higher increase in MRE11 interaction with nascent DNA (new **Fig. 8c**); we also included siRNA-mediated knockdown of MRE11 in this experiment, to further confirm the specificity of the signal.

The question remains as to what precisely the activity of PARP14 is in the context of reversed forks/ssDNA gaps – an issue that the authors allude to in the discussion section. This remains unanswered, but I think that the data provided is already mechanistic enough to be of interest to the readership of Nature Comms. Furthermore, the authors suggest (very probably correctly) that the mutation of H1682 in human PARP14 is equivalent to H1698 in murine PARP14, which was previously shown to be catalytically inactive. Nonetheless, it would complement the manuscript if the authors could attempt some validation of defective catalysis. The commercially available Af1521 macrodomain beads are excellent at pulling down both MARylated and PARylated proteins. The authors could fairly easily carry out analysis of an established PARP14 target such as STAT1 or PARP9 in macrodomain bead-mediated pulldowns from WT, PARP14 KO, PARP14 KO+ WT PARP14 and PARP14 KO+H1682 PARP14 cells, to validate that the mutant is in fact catalytically dead. In fact, these samples would be a very useful resource which the authors could employ to probe for RAD50 and other candidates that they consider potential PARP14 MARylation targets following treatment with genotoxins (though for me these latter studies extend beyond the scope of this manuscript).

We thank the reviewer for this comment. We agree that it is important to demonstrate that the H1682Q mutant is catalytically inactive. We have attempted to perform the experiment with Af1521 beads as indicated by the reviewer. However, we could not find any difference in the amount of STAT1 and PARP9 pulled down by these Af1521 beads (either agarose-crosslinked or magnetic) compared to the input, in wildtype *versus* PARP14-knockout cells (we provide below the results of these experiments as a **Figure for Reviewer 1**). This may mean that STAT1 and PARP9 are ADP-ribosylated by other ADP-ribosyltransferases than PARP14, at least in PARP14-knockout cells, or that these beads are not specifically detecting the ADP-ribosylated forms of STAT1 and PARP9. Either way, this approach does not allow us to investigate the catalytic activity of the H1682Q mutant.

Figure for Reviewer 1. Pulldowns with agarose or magnetic Af1521 beads in wildtype and PARP14-knockout HeLa cell extracts, followed by blots with anti-PARP9 or anti-STAT1 antibodies. No difference in the amount of PARP89 and STAT1 bound to these beads was observed between wildtype and PARP14-knockout cells, compared to Input.

Since we agree with the reviewer that it is important to demonstrate that the H1682Q is catalytically inactive, we decided to investigate its activity using an *in vitro* ADP-ribosylation assay with recombinant proteins. Since PARP14 is a very large protein (1801 aminoacids, 203kDa), we were unable to express in *E. coli* the full-length protein. However, a commercially-available PARP14 truncation (aa1470–1801(end)), encompassing the PARP catalytic domain has been shown to possess ADP-ribosyltransferase activity *in vitro* (BPS Bioscience, catalog #80514, <https://bpsbioscience.com/parp14-80514>). We thus decided to express in *E. coli* the same truncation, either wildtype or with the H1682Q mutation. We expressed these fragments with a 6xHis tag and purified them by NiNTA affinity. We then employed them for *in vitro* ADP-ribosylation assays using biotin-labeled NAD⁺. Since the biotin moiety is transferred to the substrate together with the ADP-ribose molecule, the ADP-ribosylated substrates can be detected by Streptavidin-HRP blots. For both the commercial (wildtype) PARP14 catalytic domain, and our own purified PARP14 wildtype catalytic domain, we noticed that incubation with biotin-NAD⁺ results in multiple bands on the Streptavidin-HRP blot, corresponding to ADP-ribosylated substrates; this signal is specifically detecting ADP-ribosylation since it did not appear in the absence of biotin-NAD⁺. These findings indicate that, in this *in vitro* assay, the catalytic domain of PARP14 is able to non-specifically ADP-ribosylate contaminating bacterial proteins from the purification of the PARP14 truncations. Importantly, the H1682Q mutant showed no Streptavidin-HRP signal, even in the presence of biotin-NAD⁺. These findings indicate that this mutant is indeed devoid of catalytic activity. We present these findings in revised manuscript as the new **Supplementary Fig. S5b,c**.

I might also suggest that in a few places, inclusion of representative images of DNA fibres, COMET tails and gammaH2AX foci might be a useful addition that would provide visual cues for the less expert reader.

In the revised manuscript, we included representative images of DNA fibers (new **Supplementary Fig. S1c**), COMET tails (new **Supplementary Fig. S2b**), and gammaH2AX foci (new **Supplementary Fig. S2a**),

The M+Ms section contains sufficient information for the data to be reproduced.

I have suggested two sets of experiments that I hope will improve the manuscript. However,

overall I consider this work suitable for publication in Nature Communications, and expect it to be of significant interest to readers with an interest in genome stability and/or cancer cell biology.

We are very happy that the reviewer found our work “suitable for publication in Nature Communications” and thank them for their constructive comments.

Reviewer #2

We are glad that the reviewer found that “this is a very elegant and original investigation” and considers our work “a significant improvement in our understanding of the mechanisms underlying reversed fork processing”. We thank the reviewer for their helpful comments, which we have addressed as indicated below.

(Remarks to the Author):

Replication fork reversal, which involves the annealing of the nascent strands of the sister chromatids, is a process whereby replication forks that have arrested due to replication stress may be processed for restart. Protection of reversed replication fork from MRE11 nuclease-driven nascent DNA degradation by the BRCA pathway is crucial to maintain genomic stability and chemoresistance. In this context, the manuscript by Dhoonmoon et al. sheds new light on the factors that are involved in this process, as well as the underlying mechanisms.

Key novelties include the identification of a novel role for the mono-ADP-ribosyltransferase PARP14 in orchestrating the engagement of the MRE11 nuclease on reversed replication forks in BRCA-deficient cells. Notably, this function requires the catalytic activity of PARP14. Specifically, loss of PARP14 suppressed MRE11-mediated fork degradation and gap accumulation, and promoted genome stability and chemoresistance in BRCA-deficient cells. In addition, the authors identify a physical interaction between PARP14 and the DNA end-binding complex KU complex and provide evidence that KU operates as a regulator of PARP14-MRE11-mediated DNA resection. They show that in human cells, KU binds the reversed fork end, as previously shown in Sc. Pombe by the Lambert's group. In addition, their data indicate that KU protects reversed replication forks from degradation mediated by EXO1. The authors integrate PARP14 as a new actor in a model that is reminiscent of the processing of single-ended DSBs resulting from the collapse of replication forks upon collision e.g. with ssDNA. In this model, Ku eviction from the DNA end generated at reversed forks is promoted by PARP14-mediated MRE11 end/exo-nuclease activities and allows long-range resection by EXO1.

This is a very elegant and original investigation that uses a broad range of cellular models generated by RNAi and CRISPR-Cas9, as well as small-molecule inhibitors, in very complementary, high-profile experiments addressing the presence and interaction of DNA repair factors at nascent DNA.

Although hundreds of PARP14 MARYlation substrates have been identified, the substrates of PARP14 in this story remain unknown. However, as the authors pointed out correctly, MARYlation modifications are hard to study. Thus, the current lack of knowledge on the targets of PARP14 during reversed fork processing should not prevent this manuscript from being considered as a significant improvement in our understanding of the mechanisms underlying reversed fork processing.

However, I would like to raise the following issues that, in my opinion, should be addressed before this manuscript is accepted.

My major concerns relate to the last experiments carried out by the authors in order to support the model presented in Figure 7H.

- Re: Fig. 7F, 7G

To support the claim that KU eviction (associated with PARP14-mediated MRE11 activity) allows EXO1 engagement and long patch fork resection, it would be important to examine KU presence by SIRF after 30 min of EdU labelling, in order to reveal a correlation between the presence of EXO1 observed at this time and loss of KU.

In the revised manuscript, we now show that inhibition of PARP14 or MRE11 significantly enhances KU SIRF foci formation after 30min of EdU labeling (new **Supplementary Fig. S6m**), in line with our model that the MRE11-PARP14 complex is required for KU eviction.

In addition, similar to what has been done for seDSBs, and to complement their experiments with a siMRE11, the authors should show that in cells where MRE11 exonuclease activity is inhibited (or better, in RNAi-mediated MRE11-depleted cells expressing an ectopic, endo/exo-defective MRE11 construct), KU is retained significantly while EXO1 is not recruited.

We thank the reviewer for this useful suggestion. Complementation studies with MRE11 endo/exonuclease mutants are technically challenging. However, to investigate the roles of MRE11 endonuclease and exonuclease activities, in the revised manuscript we have employed a specific MRE11 endonuclease inhibitor (PFM01) in addition to the exonuclease inhibitor (mirin) which we previously employed. We now show that inhibition of either the endonuclease or the exonuclease activities of MRE11 similarly result in: a) suppression of nascent DNA degradation (new **Supplementary Fig. S1a,b**), and b) suppression of EXO1 SIRF foci formation after 30min of EdU labeling (new **Fig. 8h**). These findings further validate our model that consecutive activities of MRE11 endonuclease and subsequently exonuclease activities are required to remove KU and allow EXO1 engagement, resulting in long-range nascent DNA resection.

- Re: Fig 7A-D.

From the data presented in Fig 7A-D, and the observation that KU and PARP14 interact - as assessed by PLA (Fig 6A) - the authors suggests that KU is responsible for PARP14 recruitment at reversed replication forks. However, the authors should rule out that the lack of PARP14/MRE11 recruitment observed upon loss of KU in BRCA2-deficient cells is not due to the activity of EXO1, which could prevent such recruitment (e.g. due to nucleolytic degradation) and/or displace PARP14 from the DNA. In this context, it would be important to examine whether PARP14 can be detected at reversed replication forks when BRCA2-, KU-defective cells also express a catalytically-dead EXO1 construct. This is important because the authors have previously reported data indicating an interaction between PARP14 and PCNA (Ref 19), raising the possibility that other recruitment routes to (stalled) replication forks may exist for PARP14.

During the course of the revision, we have tried to generate catalytic-dead EXO1 mutants and use them for complementation of BRCA2^{KO}EXO1^{KO} double knockout cells. However, we were not able to obtain a stable expression of the EXO1 mutant in these cells. Nevertheless, to

address the reviewer's comment, in the revised manuscript we show that KU depletion equally suppresses PARP14 and MRE11 SIRF foci formation in BRCA2^{KO} single knockout cells and BRCA2^{KO}EXO1^{KO} double knockout cells (new **Supplementary Fig. S6j,k**), thus ruling out an involvement of EXO1 activity in the suppression of the recruitment of the PARP14-MRE11 complex upon KU depletion.

Other points:

- INTRODUCTION (last sentence of page 3): "Specific inhibitors of PARP14 catalytic activity have recently been developed, and targeting PARP14 has been proposed as a possible therapeutic approach for multiple cancer types (10, 13, 14, 16-18)."
Please specify some of the reasons why PARP14 is seen as a relevant therapeutic target.

In the revised manuscript, we have listed pro-cancer activities reported for PARP14 in the manuscripts we cited (page 4 in the revised manuscript).

RESULTS

- Fig. 1D: The authors should examine the impact of co-depleting ZRANB3 and PARP14, as this may reveal additional roles for PARP14 upstream from fork degradation. In addition, in this experiment, cells with siRNA-PARP14 behave like siCTL cells. How consistent is this with previous findings (Ref 19) by the authors regarding the hypersensitivity of PARP14-depleted cells to HU and the accumulation of gH2AX foci in these cells?

This particular DNA fiber experiment (Fig. 1d in the revised manuscript) is designed to measure fork slowing upon treatment with low-dose HU for 2h, a surrogate readout for fork reversal. As pointed out by the reviewer, PARP14 depletion has no impact on replication fork tracts under these conditions, indicating that it is not involved in fork slowing/reversal. We do not think that these findings are in contrast with our previous findings that loss of PARP14 results in mild sensitivity to HU, since the experimental conditions in this particular assay (low-dose HU for a short time) are designed to measure specifically only fork slowing, and not other processes that may affect HU sensitivity.

In the revised manuscript, we performed the experiment requested by the reviewer and show that co-depletion of PARP14 and ZRANB3 causes a similar change in replication tracts ratio as ZRANB3 depletion by itself (new **Supplementary Fig. S1j**). These findings are in line with our model that, unlike ZRANB3, PARP14 is not involved in fork slowing or reversal.

- Fig. 2A: What concentration of HU was used in this experiment with DLD1 cells? How long were the cells exposed to HU?

We apologize for erroneously omitting to provide this information in the original submission. In the revised manuscript, we now provide this information in the legend for Fig. 2a.

- Fig. 2F: In their previous ref 19, the authors showed that exposure of HeLa cells to 0.05 microM Olaparib resulted in a clonogenic survival of about 40% (siCTL) and 20% (siPARP14-1). Here, the clonogenic survival are much higher, even at much higher doses of Olaparib. Why is this?

The differences indicated by the reviewer are caused by the different experimental setups employed (as reported in the respective Methods sections). Briefly, in the current manuscript, the cells were incubated with drugs for 3 days only, after which the media was replaced with fresh media. In our previous publication (ref. 19), cells were incubated with the drugs for 2 weeks.

- Fig. S3A-B: Can the authors report the statistical significance of other relevant comparisons? Indeed, it looks like the impact of depleting PARP14, BRCA1 and BRCA2 is similar in these cells.

We added the statistical analyses requested by the reviewer. As we previously reported (ref. 19), the impact of PARP14 depletion on homologous recombination, as measured by the DR-GFP assay, is significant but generally milder than the impact of BRCA1/2 depletion. In the particular experiment presented in this figure, we employed a lower amount of BRCA1 or BRCA2 siRNA since we noticed that these siRNAs can be toxic in these cells, particularly in combination with other siRNAs (for co-depletion experiments). This may explain why depletion of BRCA1 or BRCA2 does not show a stronger effect on HR in this assay, as would have been anticipated.

- Fig. 3C: I would recommend showing images of the biotin-MRE11 foci without DAPI counterstaining but with nuclear contours instead, to facilitate their visualization. This should also apply to other figures showing DAPI-counterstained nuclei (e.g. Fig 4A, 4C, 6A).

In the revised manuscript, we now present the SIRF images as requested by the reviewer (new **Fig. 3e, 4a, 4c, 7a**), and moved the original SIRF micrographs to the Supplementary Material (**Supplementary Fig. S4a, S4c, S4f, S6a**).

- Replace "exogenous" by "ectopic" in this sentence: "we noticed that stable exogenous re-expression of wildtype".

We made the text change requested by the reviewer (page 14).

- Fig. 5B-G: the spacing in the titles of these figures should be changed to increase readability.

We made the changes requested by the reviewer.

- In the following sentence the authors should also provide the underlying mechanism of the PARP14 inhibitors: "Specific PARP14 inhibitors have been recently developed (16, 64) and are commercially available."

We provided this information in the revised manuscript (page 16).

- Fig. 5O. In this figure, for both inhibitors, it appears that the viability of cells exposed to the inhibitor or the combination [inhibitor + cisplatin] is almost identical in WT and mutant cells. The only difference here is the initial sensitivity of WT and BRCA2 KO cells to cisplatin. The impact

of the inhibitors alone on cell viability is identical in WT and mutant cells. As this assay was carried out using a single concentration of cisplatin, and in order to control that the results do not reflect limitations of the ATP-dependent cell viability assay, I would recommend to re-do this experiment using clonogenicity assays over multiple cisplatin concentration.

As requested by the reviewer, we now performed the experiment using clonogenic assays and multiple cisplatin concentration. We show that the two PARP14 inhibitors (H10 and RBN012759) do indeed rescue the cisplatin sensitivity of BRCA2-knockout cells (new **Fig. 6g,h**).

- Fig. 7C: Can the authors comment on the MRE11 SIRF data in HeLa cells versus DLD1 cells. MRE11 binding to nascent DNA seems to be less impacted in HeLa compared to DLD1 cells. Does this suggest other recruitment mechanisms in these cells? The authors should use a siMRE11 control in 7C, similar to the siExo control of 7E.

While the trend was always the same, we did notice some variation in the fold increase in MRE11 SIRF foci from experiment to experiment. We do not think it is cell line specific as we did not notice significant differences between cell line. In fact, this experimental variation was observed by other scientists in the field as well (see above comment by Reviewer #1). In the revised manuscript, we present the results of a replicate experiment in which the fold increase is higher (new **Fig. 8c**). As requested by the reviewer, we also include siRNA targeting MRE11 in this experiment, to demonstrate the specificity of the signal.

- Add "nascent" in the following sentence as such: We reasoned that EXO1 engagement on "nascent" DNA in KU-proficient cells takes place at a later stage in fork degradation.

We thank the reviewer for pointing out the missing word; we made the correction in the revised manuscript (page 19).

Reviewer #3

We are glad that the reviewer found that "this work identifies an important role for PARP14", "is well written and clearly presented", "the conclusion drawn are appropriate", and "the experiments are well designed and executed". We thank the reviewer for their helpful comments, which we have addressed as indicated below.

(Remarks to the Author):

In this manuscript Dhoonmoon et al. assess the role of PARP14 in replication fork protection and processing. They propose that PARP14 targets Mre11 to stalled replication forks and as a consequence, loss of PARP14 suppresses Mre11-dependent degradation of replication forks in BRCA-deficient cells. These events are dependent on PARP14 catalytic activity, which is proposed to regulate the process by promoting accumulation of PARP14 at sites of replication stress. They also test the role of Ku70/80 in these events and propose that Ku targets PARP14/Mre11 to regressed forks, allowing partial resection that releases Ku to subsequently promote Exo1-dependent DNA resection.

The mechanism of processing replication forks and how this contributes to genome stability and DNA damage tolerance is topical. This work identifies an important role for PARP14 in this process, which is novel. The manuscript is well written and clearly presented. Generally the conclusion drawn are appropriate. As a whole, the experiments are well designed and executed. However, there are areas where additional controls would strengthen the study and these have been highlighted below.

Specific comments:

Data in figure 1A needs to be assessed using siRNA of Mre11 as opposed to Mirin. This will also offer the opportunity to assess whether Mre11 and PARP14 are epistatic with regards suppressing replication fork degradation in BRCA-deficient cells. This is a critical experiment to place Mre11 and PARP14 in the same pathway, as suggested here and throughout the manuscript. Similar to observations in Figs 1E and F, these experiments should be repeated in other cell lines.

For the revised manuscript, we performed the PARP14 and MRE11 co-depletion in multiple BRCA-deficient cell lines (HeLa-BRCA2^{KO}, DLD1-BRCA2^{KO}, RPE1-BRCA1^{KO}) as requested by the reviewer (new **Supplementary Fig. S1e,k,l**). Co-depletion of PARP14 and MRE11 did not change the replication tracts ratios compared to individual depletions, in line with our model that PARP14 and MRE11 act in the same pathway (although we would like to point out that the individual depletions of PARP14 or MRE11 already fully restore fork protection since the ratios are around 1, and thus the co-depletion would anyway not be expected to further enhance the phenotype)

Fig 1D: Fork slowing is only a surrogate marker of fork reversal and could be caused by a variety of factors. An alternative needs to be used to assess fork reversal (e.g. electron microscopy or recognition of forks by GFP-RuvA; www.nature.com/articles/s41467-019-09196-9).

We agree that it is important to validate the fork slowing experiments with other readouts of fork reversal. We thank the reviewer for suggesting these assays. Electron microscopy is technically very challenging, requiring special instrumentation and expertise, and was only performed so far in a small number of laboratories in the world. Regarding the RuvA assay, we were not able to find the proper experimental conditions to employ it reliably, within the time frame of this submission. Nevertheless, in the revised manuscript we are addressing the reviewer's comment using a different approach: We employed the SIRF assay to measure PARP1 engagement on stalled replication forks. Since PARP1 is a critical regulator of fork reversal (PMID: 23396353), the presence of PARP1 on nascent DNA upon replication stress has been previously used as a readout for fork reversal (PMID: 31255466). Unlike ZRANB3 depletion, which as expected reduced PARP1 SIRF foci, PARP14 knockdown did not affect it (new **Fig. 1e**). These results are in line with the fiber slowing experiments, and further validate our model that PARP14 does not participate in fork reversal.

Is restoration of replication fork degradation by expression of PARP14 dependent on the catalytic or DNA binding activities? Catalytic activity is tested later, but could a DNA binding domain mutant of PARP14 also be used to test if it is no longer able to restore replication fork

degradation in BRCA2 cells? This might also help assess separation of function of PARP14 catalytic activity and DNA binding.

We agree with the reviewer that investigating PARP14 DNA binding activities is very important. However, as of now, it is not known how PARP14 binds DNA. Indeed, other than the SIRF studies presented in our manuscript, we are not aware of previously-published experiments showing PARP14 binding to DNA. While we speculate in the Discussion section that PARP14 may interact with DNA via its two N-terminal RRM domains (since the RRM domain fold was previously shown to bind ssDNA), we have no experimental indication that this may truly be the case. Validation of RRM as the DNA-binding domain, or identification of the real DNA binding region if it is not RRM, and subsequent generation of mutants which are unable to bind DNA, is complicated and time-intensive, and thus not feasible within the timeline of this revision. We would like to respectfully argue that these studies are outside the scope of the current manuscript, but they represent the future focus of studies in our laboratory.

During the course of this revision, we attempted to investigate the relevance of the RRM domains by overexpressing a truncation (aa1-300) of PARP14 encompassing only these domains. We reasoned that, if this region is involved in PARP14 recruitment to nascent DNA, its overexpression may act in a dominant negative manner and interfere with the recruitment of endogenous wildtype PARP14, as well as its binding partner MRE11. HeLa cells with inducible expression of this truncation were subjected to PARP14 and MRE11 SIRF assays following BRCA2 knockdown. While the induction of PARP14 and ME11 SIRF foci upon BRCA2 knockdown was still observable in the RRM-overexpressing cells, it seemed to be attenuated compared to control cells. We provide these results as the **Figure for Reviewer 3** below. These findings suggest a potential role for the RRM domains in nascent DNA binding. Future experiments in our laboratory will explore this possibility, but as mentioned above, we consider this to be beyond the scope of the current manuscript,

As a whole, the data in Fig 2 support the proposed hypothesis – that PARP14 disruption suppresses DNA damage and sensitivity of BRCA-deficient cells to cisplatin/olaparib. However, there are some experimental design issues and inconsistencies in data that need to be addressed. Fig 2 J&K: Parental cells and PARP14 KO cells need to be included in this analysis. Without these controls the experiment is difficult to interpret.

For the revised manuscript, we performed these experiments including the wildtype and PARP14-knockout cells (new **Supplementary Fig. S2c,d**).

Fig 2K: PARP14 KO#2 data, whilst statistically significant, is not dramatic.

We repeated this experiment and provide the results in the new **Fig. 2k**. While still not dramatic, the rescue is more clear in this experiment.

Fig 2L: The rescue of olaparib sensitivity with PARP14 KO is not dramatic.

We agree that the result is not dramatic in this cell line, which may reflect incomplete BRCA depletion (since in these cell lines we use siRNA to knock-down BRCA expression, as opposed to the BRCA-knockout cell lines used in the other experiments). Nevertheless, we believe that overall our results are consistent through multiple cell lines, with multiple gene inactivation approaches.

Untreated cells need to be included in Supp Fig, S2.

In this figure (Supplementary Fig. S2g in the revised manuscript), the results are shown as normalized to untreated cells (which is why this condition is not shown, since it would be equal to 1). We are now clearly indicating this in the figure legend.

Using the DR-GFP assay to measure HR efficiency indicate that chemoresistance upon PARP14 loss is not due to restoration of HR. However, this assay is not measuring HR in the context of replication stress, but at a Isc1-induced DSB. It would be more informative to look at HR events associated with S-phase, such as sister chromatid exchanges, for example.

We agree with the reviewer that the DR-GFP assay, just like any other reporter assay, has limitations. Unfortunately, cytological sister chromatid exchange assays are tedious and impractical to establish in our laboratory during the revision timeline. However, we would like to point out that I-SceI-induced recombination is considered a sister chromatid exchange assay (PMID: 10880452). Indeed, HR as measured by the DR-GFP assay was shown to require ATR and not ATM (PMID: 17412408) suggesting that this reporter assay measures replication-associated recombination, rather than the “classic” recombination that takes place in G2 at double stranded breaks.

We would also like to point out that in our original manuscript, in addition to the DR-GFP assay, we also showed that PARP14 loss does not restore RAD51 levels upon camptothecin treatment in BRCA-deficient cells. Since camptothecin induces replication-dependent double strand breaks, this result provides yet another indication that PARP14 loss does not restore replication-associated recombination in BRCA-deficient cells. In the revised manuscript, we expanded this experiment to include multiple PARP14 siRNA oligonucleotides, and additional control conditions (new **Supplementary Fig. S3e**). Ultimately, even if PARP14 may have some

impact on replication-associated recombination (which we now acknowledge as a possibility in the Discussion section of the revised manuscript -page 20), our work clearly indicates that PARP14 has a novel function in MRE11-mediated fork degradation and gap suppression, which we think represents the main conceptual advance generated by our study.

Fig 3: The suppression of Gap accumulation on PARP14 loss needs to be assessed in cell types other than HeLa. Given Mre11 also suppresses gap accumulation, it should be tested whether PARP14 and Mre11 are epistatic with regards this phenotype. This is critical to support the hypothesis that they function in the same pathway, as suggested in Fig 2D-H and subsequent figs in the manuscript.

In the revised manuscript, we extend this analyses in RPE1-BRCA1^{KO} cells (new **Fig. 3b**). Moreover, as requested by the reviewer, we show that in both HeLa-BRCA2^{KO} cells and RPE1-BRCA1^{KO} cells, co-depletion of PARP14 and MRE11 results in similar suppression of comet tail moment as the single knockdowns (new **Fig. 3b**, new **Supplementary Fig. S3f, S1f**), suggesting that they are epistatic and act in the same pathway.

PLA does not necessarily indicate 2 proteins interact, just that they get to similar regions of the genome. A co-IP needs to be performed to say the 2 proteins interact.

In the revised manuscript, we now show that PARP14 and MRE11 interact in co-immunoprecipitation experiments (new **Supplementary Fig. S4d**).

Controls of Mre11 and PARP14 antibodies alone need to be included in Fig 4B. These controls (antibodies alone) should be included in all PLA throughout the manuscript.

In the revised manuscript, we include controls of PARP14 and MRE11 antibodies alone, as well as single antibody controls for the other antibodies used for PLA or SIRF experiments, namely KU70, KU80 and EXO1 (new **Supplementary Fig. S4a, S4e, S6b, S6c, S6I**). We would also like to point out that all our PLA experiments are additionally controlled by including cells with knockdown or knockout of the proteins investigated, which validates the specificity of the signal.

Fig 4F: Parp1 has also been implicated in maintaining reversed forks by inhibiting RecQ, so the siPARP1 phenotype is not necessarily due to fork slowing.

We agree with the reviewer that PARP1 has multiple roles in fork protection, and it is difficult to tease out which of those may be responsible for the observed impact on PARP14 recruitment. Nevertheless, the ZRANB3 data presented in the same experiment clearly indicates that fork reversal is important for PARP14 recruitment.

Fig 5A-G: Is there a reason different cells are shown in different panels? Are these independent experiments? It would be more appropriate to compare cell types within the same experiments and this should be performed.

These figures were in fact derived from the same experiments: Figs 5b,c are PARP14 SIRF results from the same experiment, and Figs 5d-g are MRE11 SIRF results from the same

experiment. We have kept the Y-axis scale identical for the results derived from the same experiment, so that the cell lines can be compared, as mentioned by the reviewer. The only reason we are showing the results in separate panels is that we believe this makes this complicated figure easier to understand by the reader.

The PARP14 H1682Q mutant should be tested for a lack of catalytic activity in vitro.

As we also indicate in the response to Reviewer #1 above, we agree with the reviewers that it is important to demonstrate that the H1682Q is catalytically inactive. We thus decided to investigate its activity using an *in vitro* ADP-ribosylation assay with recombinant proteins. Since PARP14 is a very large protein (1801 aminoacids, 203kDa), we were unable to express in *E. coli* the full-length protein. However, a commercially-available PARP14 truncation (aa1470–1801(end)), encompassing the PARP catalytic domain has been shown to possess ADP-ribosyltransferase activity *in vitro* (BPS Bioscience, catalog #80514, <https://bpsbioscience.com/parp14-80514>). We thus decided to express in *E. coli* the same truncation, either wildtype or with the H1682Q mutation. We expressed these fragments with a 6xHis tag and purified them by NiNTA affinity. We then employed them for *in vitro* ADP-ribosylation assays using biotin-labeled NAD⁺. Since the biotin moiety is transferred to the substrate together with the ADP-ribose molecule, the ADP-ribosylated substrates can be detected by Streptavidin-HRP blots. For both the commercial (wildtype) PARP14 catalytic domain, and our own purified PARP14 wildtype catalytic domain, we noticed that incubation with biotin-NAD⁺ results in multiple bands on the Streptavidin-HRP blot, corresponding to ADP-ribosylated substrates; this signal is specifically detecting ADP-ribosylation since it did not appear in the absence of biotin-NAD⁺. These findings indicate that, in this *in vitro* assay, the catalytic domain of PARP14 is able to non-specifically ADP-ribosylate contaminating bacterial proteins from the purification of the PARP14 truncations. Importantly, the H1682Q mutant showed no Streptavidin-HRP signal, even in the presence of biotin-NAD⁺. These findings indicate that this mutant is indeed devoid of catalytic activity. We present these findings in revised manuscript as the new **Supplementary Fig. S5b,c**.

Additionally, a potential explanation for the data presented is that the H1682Q mutation impacts on the structure of PARP14 to disrupt DNA binding activity independently of catalytic activity. Do PARP14 inhibitors similarly prevent interaction of PARP14 with nascent DNA and/or PLA with Mre11? This is straightforward and should be tested. Can the DNA binding activity of PARP14 H1682Q be tested in vitro? Are there other mutations that can be exploited to separate the functions of catalytic inactivation and DNA binding? It is important that these possibilities are tested.

We thank the reviewer for these very useful suggestions. In the revised manuscript, we now show that PARP14 inhibitors suppress PARP14 SIRF foci formation (new **Fig. 6f**), indicating that the catalytic activity is required for PARP14 recruitment to nascent DNA and arguing that the SIRF results obtained with the H1682Q mutant are not caused by structural defects of this mutant.

As described above, to our knowledge the PARP14 DNA binding activity has not been previously characterized *in vitro*, and no other PARP14 mutants have been described to separate the catalytic function from the DNA binding activity. Moreover, the full PARP14 protein is too large to be easily purified. These experiments are unfeasible within the timeline of this revision, and are outside the scope of our current submission, but represent the focus of our future studies. While we agree that it is important to study in the future the impact of DNA

binding domain mutation, our current work focuses on the catalytic function. In this context, our experiments with both the PARP14 catalytic inhibitors and the H1682Q catalytic mutant clearly show that PARP14 catalytic activity regulates its recruitment to nascent DNA, as well as MRE11-mediated nascent DNA degradation and ssDNA gap expansion.

Parp14 KO cells should be included in Fig 5J.

The PARP14^{KO} cells are shown in Fig. 3c.

The manuscript would be strengthened by inclusion of data testing the importance of PARP14 catalytic activity in promoting fork degradation in other cell lines. Given the availability of PARP14 inhibitors, this should be straightforward.

We thank the reviewer for this suggestion. In the revised manuscript, we show that the inhibitors also rescue fork degradation in BRCA1-knockout RPE1 cells and in BRCA2-knockout DLD1 cells (new **Fig. 6b,c**). Moreover, we show that the inhibitors also suppress HU-induced ssDNA gap accumulation in BRCA2-knockout cells (new **Fig. 6e**).

Fig 6: For PLA experiments, controls need to be included using PARP14 and Ku80 antibodies alone.

As also mentioned in the response to the reviewer's comment above, in the revised manuscript we include controls for all antibodies used for PLA or SIF experiments, including PARP14 and KU80 (new **Supplementary Fig. S4a, S4e, S6b, S6c, S6l**).

The data comparing WT and PARP14 KO cells need to be quantified.

As there are no PARP14 PLA foci in PARP14^{KO} cells, this quantification was not included in the figure (since it would basically be 0).

The suppression of Ku SIF signal by depletion of ZRANB3 is not dramatic for Ku70. Can this be improved by depleting other fork regression factors (e.g. SMARCAL1, PARP1 etc.) alone, or in combinations?

In the revised manuscript, we now show that depletion of ZRANB3, SMARCAL1, or PARP1 can equally reduce KU SIF foci formation (new **Fig. 7f**, new **Supplementary Fig. S6e**). We have attempted the co-depletion experiments suggested by the reviewer, but in our hands, this caused significant cell death and we were not able to obtain robust results. Nevertheless, we would like to point out that previous studies have shown that individual depletions of SMARCAL1 or ZRANB3 completely suppressed fork degradation in BRCA-deficient cells, indicating that they act in an epistatic manner in fork reversal, and thus an additive phenotype is not expected.

REVIEWERS' COMMENTS

Reviewer #1 (Remarks to the Author):

Dhoonmoon and colleagues have submitted an extensively modified manuscript, which contains additional data addressing my initial minor concerns. I will not deal with the additional work addressing concerns raised by other reviewers, but suffice it to say that I am relatively satisfied with the work carried out following the first review.

The authors have conducted additional repeats to address some inconsistencies in the magnitude and consistency of certain phenotypes observed between experiments in the original manuscript. The data now appears improved in terms of consistency between experiments.

The authors also carried out appropriate attempts to perform some of the experiments I recommended in the last review round. In particular, they have performed experiments to attempt to validate the assumption that the mutant of PARP14 employed here is catalytically inactive. There were a few issues with non-specific ADP-ribosylation and likely cross-ribosylation of PARP14 substrates by other PARPs in the same. Nonetheless, overall their data does indeed suggest that the H1682 mutant lacks detectable catalytic activity. As such, all claims associated with this mutant in the original manuscript are perfectly valid.

The authors also made a number of minor changes to the presentation of the manuscript itself based on my suggestions. I am happy with these modifications.

Overall, my assessment is that this manuscript (which was already reasonably sound) has been significantly improved by the authors efforts based on the initial round of reviews.

Reviewer #2 (Remarks to the Author):

The KU-PARP14 axis differentially regulates DNA resection at stalled replication forks by MRE11 and EXO1.

I have now read the revised manuscript, as well as the responses provided by Dhoonmoon and colleagues to the comments of all three reviewers.

I am satisfied with the way the authors have addressed these comments. In particular, I think it was important to provide some confirmation that H1682Q mutation abolishes the catalytic activity of human PARP14. In addition, the additional experiments linking KU eviction to PARP14-mediated MRE11 activity now provide strong support for the model presented in Fig 7H. I also think the inclusion of representative images obtained during DNA fiber assays, as well as COMET assays and DSB foci analysis, which was done at the suggestion of Reviewer 1, will facilitate the reading by non-experts.

This manuscript represents an original and highly significant contribution to our knowledge of the factors and mechanisms governing replication fork stability. I therefore recommend it for publication in Nature Communications.

Reviewer #3 (Remarks to the Author):

The authors have addressed the vast majority of my concerns and this is now a vastly improved

manuscript. I only have two minor comments before publication, both of which I hope can be resolved easily:

a) My previous comments suggested co-depletion of PARP14 and Mre11 to test whether they are epistatic in the context of suppressing gap formation in BRCA2 KO cells. The authors have included these data and this is clearly the case in RPE1 cells (Fig3b). However, the co-depletion is additive in HeLa (Supplementary Figure S3f). The authors present a wealth of data providing a link between PARP14 and Mre11 in this study, so this should not be a sticking point. However, given these data have been presented, I feel it is appropriate to acknowledge them in the manuscript and perhaps include a brief caveat/explanation for this inconsistency.

b) Molecular weight markers should be included in western blots.

Response to referees

We are happy that the reviewers found that our revised manuscript satisfactorily addressed their comments, and recommended the manuscript for publication. Below, please find our responses to the reviewer's remaining comments (**our responses in red font**).

Reviewer #1

Dhoonmoon and colleagues have submitted an extensively modified manuscript, which contains additional data addressing my initial minor concerns. I will not deal with the additional work addressing concerns raised by other reviewers, but suffice it to say that I am relatively satisfied with the work carried out following the first review.

The authors have conducted additional repeats to address some inconsistencies in the magnitude and consistency of certain phenotypes observed between experiments in the original manuscript. The data now appears improved in terms of consistency between experiments.

The authors also carried out appropriate attempts to perform some of the experiments I recommended in the last review round. In particular, they have performed experiments to attempt to validate the assumption that the mutant of PARP14 employed here is catalytically inactive. There were a few issues with non-specific ADP-ribosylation and likely cross-ribosylation of PARP14 substrates by other PARPs in the same. Nonetheless, overall their data does indeed suggest that the H1682 mutant lacks detectable catalytic activity. As such, all claims associated with this mutant in the original manuscript are perfectly valid.

The authors also made a number of minor changes to the presentation of the manuscript itself based on my suggestions. I am happy with these modifications.

Overall, my assessment is that this manuscript (which was already reasonably sound) has been significantly improved by the authors efforts based on the initial round of reviews.

We thank the reviewer for their kind comments, and are happy to learn that the reviewer found that our revised manuscript was significantly improved, and addressed their concerns.

Reviewer #2

I have now read the revised manuscript, as well as the responses provided by Dhoonmoon and colleagues to the comments of all three reviewers.

I am satisfied with the way the authors have addressed these comments. In particular, I think it was important to provide some confirmation that H1682Q mutation abolishes the catalytic activity of human PARP14. In addition, the additional experiments linking KU eviction to PARP14-mediated MRE11 activity now provide strong support for the model presented in Fig 7H. I also think the inclusion of representative images obtained during DNA fiber assays, as well as COMET assays and DSB foci analysis, which was done at the suggestion of Reviewer 1, will facilitate the reading by non-experts.

This manuscript represents an original and highly significant contribution to our knowledge of the factors and mechanisms governing replication fork stability. I therefore recommend it for

publication in NatureCommunications.

We thank the reviewer for their kind comments, and are happy to learn that the reviewer found that our revised manuscript addressed their concerns, and recommended its acceptance.

Reviewer #3

The authors have addressed the vast majority of my concerns and this is now a vastly improved manuscript. I only have two minor comments before publication, both of which I hope can be resolved easily:

We thank the reviewer for their kind comments, and are happy to learn that the reviewer found that our revised manuscript was vastly improved. Below, we are addressing the remaining comments of the reviewer.

a) My previous comments suggested co-depletion of PARP14 and Mre11 to test whether they are epistatic in the context of suppressing gap formation in BRCA2 KO cells. The authors have included these data and this is clearly the case in RPE1 cells (Fig3b). However, the co-depletion is additive in HeLa (Supplementary Figure S3f). The authors present a wealth of data providing a link between PARP14 and Mre11 in this study, so this should not be a sticking point. However, given these data have been presented, I feel it is appropriate to acknowledge them in the manuscript and perhaps include a brief caveat/explanation for this inconsistency.

We acknowledge this issue in the revised manuscript, and suggest that this could potentially be caused by cell line differences in the case of this particular assay (page 11).

b) Molecular weight markers should be included in western blots.

We included molecular weight markers in the western blots (Fig. 3i and Fig. 5a)